# A single cell atlas of the human liver tumor microenvironment

Hassan Massalha[1] (ID), Keren Bahar Halpern[1], Samir Abu-Gazala[2,3], Tamar Jana[1], Efi E Massasa[1], Andreas E Moor[4] (ID), Lisa Buchauer[1] (ID), Milena Rozenberg[1], Eli Pikarsky[5], Ido Amit[6], Gideon Zamir[2] & Shalev Itzkovitz[1,*] (ID)

## Abstract

Malignant cell growth is fueled by interactions between tumor cells and the stromal cells composing the tumor microenvironment. The human liver is a major site of tumors and metastases, but molecular identities and intercellular interactions of different cell types have not been resolved in these pathologies. Here, we apply single cell RNA-sequencing and spatial analysis of malignant and adjacent non-malignant liver tissues from five patients with cholangiocarcinoma or liver metastases. We find that stromal cells exhibit recurring, patient-independent expression programs, and reconstruct a ligand–receptor map that highlights recurring tumor–stroma interactions. By combining transcriptomics of laser-capture microdissected regions, we reconstruct a zonation atlas of hepatocytes in the non-malignant sites and characterize the spatial distribution of each cell type across the tumor microenvironment. Our analysis provides a resource for understanding human liver malignancies and may expose potential points of interventions.

**Keywords** human cell atlas; liver cancer; single cell RNAseq; spatial transcriptomics; tumor-stroma interactions
**Subject Categories** Cancer; Methods & Resources
**Mol Syst Biol.** (2020) 16: e9682

## Introduction

Cancer is a heterogeneous disease, exhibiting both interpatient and intrapatient variability (Marusyk et al, 2012; Meacham & Morrison, 2013; Patel et al, 2014; Alizadeh et al, 2015). Tumor cells do not operate in isolation, but rather closely interact with a complex milieu of supporting stromal cells that form the tumor microenvironment (TME) (Polyak et al, 2009; Hanahan & Weinberg, 2011; Lambrechts et al, 2018). These cells include, among others, a range of immune cells, cancer-associated fibroblasts (CAFs), and endothelial cells. Interactions between the tumor and stromal cells are critical for cancer cell survival (Meacham & Morrison, 2013). Stromal cells supply the cancer cells with growth factors, facilitate immune evasion, and modulate the composition of the extracellular matrix. Given the diversity of cell types that form the TME, it is essential to apply single cell approaches to resolve their molecular identities (Tirosh et al, 2016; Puram et al, 2017; Lambrechts et al, 2018).

The liver is a major site of both primary tumors and metastases (Llovet et al, 2016). Tumors of liver origin include hepatocellular carcinomas (Guichard et al, 2012), cholangiocarcinomas [tumors originating from liver cholangiocytes (Patel, 2011; Sia et al, 2013)], and hepatoblastomas. Liver metastases often originate in colorectal and pancreatic tumors and are the main cause of mortality in these cancer patients (Weinberg, 2013). Single cell atlases have provided important insight into the development (Camp et al, 2017; Segal et al, 2019; Popescu et al, 2019), physiology (MacParland et al, 2018; Aizarani et al, 2019), and pathology (Zhang et al, 2019, 2020; Ramachandran et al, 2019; Sharma et al, 2020) of the human liver. Here, we reconstruct a cell atlas of the malignant human liver in patients with liver metastases or cholangiocarcinomas. Our analysis highlights recurring stromal cell type signatures and interaction modalities with the carcinoma cells. By combining spatial information, we reconstruct zonation patterns of hepatocytes in the non-malignant tissue sites and identify distinct spatial distributions of cell types across the TME.

## Results

### A cell atlas of the human liver tumor microenvironment

To assemble a cell atlas of the human liver TME, we analyzed tissues from six patients who underwent liver resection (Fig 1A, Appendix Fig S1). Three Patients underwent hepatic resection for

---

1   Department of Molecular Cell Biology, Weizmann Institute of Science, Rehovot, Israel
2   Department of General Surgery, Hadassah Hebrew University Medical Center, Jerusalem, Israel
3   Transplant Division, Department of Surgery, Hospital of the University of Pennsylvania, Philadelphia, PA, USA
4   Department of Biosystems Science and Engineering, ETH Zürich, Basel, Switzerland
5   The Lautenberg Center for Immunology, Institute for Medical Research Israel-Canada, Hebrew University Medical School, Jerusalem, Israel
6   Department of Immunology, Weizmann Institute of Science, Rehovot, Israel
    *Corresponding author. Tel: +972 89343104; E-mail: shalev.itzkovitz@weizmann.ac.il

colorectal metastases, two for intrahepatic cholangiocarcinoma, and one for a cyst at a benign stage (Dataset EV1). We dissociated the tissues into single cells and measured their transcriptomes using MARS-seq (Jaitin et al, 2014; Materials and Methods). In parallel, we preserved tissues for spatial analysis using laser-capture microdissection (LCM) (Moor et al, 2017, 2018) and single molecule fluorescence in situ hybridization (smFISH) (Bahar Halpern et al, 2015).

Our single cell atlas included 7,947 cells, 4,140 from the malignant sites and 3,807 from the non-malignant sites (Fig 1B). The non-malignant sites did not show histological signs of fibrosis, with the exception of the cholangiocarcinoma patient p2 (Materials and Methods, Dataset EV1). The cells formed 17 clusters, which we annotated based on known marker genes and a recent cell atlas of cirrhotic human livers (Ramachandran et al, 2019) (Fig 1C). Notably, the stromal clusters included a mixture of cells from different patients (Appendix Fig S1), demonstrating recurring stromal signatures. Cells from the non-malignant liver sites included clusters of hepatocytes and several non-parenchymal cell populations— hepatic stellate cells, vascular smooth muscle cells (vSMC), Kupffer cells, T cells, B cells, liver sinusoidal endothelial cells (LSEC), liver vascular endothelial cells (LVEC), and cholangiocytes, the latter clustering with the carcinoma cells. Cells from the malignant liver sites included carcinoma cells, marked by KRT8, KRT18, and EPCAM (Puram et al, 2017; Fig 1B) and diverse TME cell populations, including fibroblasts, endothelial cells, and immune cells (Fig 1C). Carcinoma cells exhibited distinct gene expression differences between the cholangiocarcinoma patients and the metastatic patients (Appendix Fig S1E). Genes elevated in cholangiocarcinomas included higher expression of the cholangiocyte gene Beta-defensin 1 (DEFB1) (Harada et al, 2004) and FGFR2. Genes elevated in colorectal cancer metastasis included higher expression of Cadherin 17 (CDH17) (Panarelli et al, 2012) and the adhesion molecules CEACAM5 and CEACAM6, previously shown to correlate with metastasis colonization (Powell et al, 2018). We extracted global gene expression signatures and unique markers for each of these cell types (Fig 1D, Datasets EV2 and EV3). We validated the expression of a panel of 12 marker genes using smFISH (Appendix Fig S2).

## TME cell types exhibit recurring expression signatures

A common question in single cell analysis is whether the reconstructed cell atlases are stable with regard to the numbers of cells per sample and the numbers of samples (Mereu et al, 2020). This question is particularly important in cancer, due to the profound levels of interpatient heterogeneity (Marusyk et al, 2012;

Meacham & Morrison, 2013; Patel et al, 2014; Alizadeh et al, 2015). We assessed the stability of the expression signatures obtained from our atlas with regard to the number of sampled patients and the number of sampled cells. To this end, we reconstructed the mean gene expression signatures for each of the 17 cell type clusters, based on subsamples of the six patients, and equally sized subsamples of cells from all patients as controls. We compared these mean expression signatures of subsets of the data with those obtained from the full atlas. We found that the gain in correlations, when adding new patients, strongly curtailed for most cell types beyond three patients and converged on the correlations obtained when subsampling cells rather than patients (Appendix Fig S3). An exception was the carcinoma cluster, where gene expression signatures changed with each new added patient (Appendix Fig S3). Our analysis thus demonstrates that, while carcinoma cells exhibit high interpatient variability, the liver TME exhibits recurring gene expression signatures that are more uniform between patients.

## Differences in TME gene expression between the malignant and non-malignant sites

Our single cell analysis of matching malignant and non-malignant sites within the same patients enabled identification of gene expression differences in distinct cell populations that compose the TME (Fig 2). Genes elevated in tumor endothelial cells compared to the non-tumor endothelial cells included the von Willebrand factor VWA1, encoding a glycoprotein previously shown to facilitate tumor cell extravasation (Terraube et al, 2007), as well as SOX17 (Yang et al, 2012) and INSR (Nowak-Sliwinska et al, 2019), both shown to promote tumor angiogenesis (Fig 2A). The immune cell populations in the malignant liver predominantly included scar-associated macrophages (SAMs) (Ramachandran et al, 2019; Fig 2 B). These cells express the marker genes CD9 and TREM2, a tumor suppressor in hepatocellular carcinoma (Tang et al, 2019), as well as the markers CAPG and GPNMB. GSEA analysis of Subramanian et al (2005) SAM genes resulted in a significant enrichment of apical junction genes and the complement system. Their recurring signatures included lipid-associated genes, such as PLIN2 and LPL, overlapping the recently identified SPP1[+] lipid-associated macrophages (LAMs) in mouse fatty livers (Remmerie et al, 2020). Liver mononuclear phagocyte populations from the non-malignant liver sites were composed of Kupffer cells, expressing C1QB, MARCO, CD5L, and CD163 (Appendix Fig S4). T cells from the malignant sites were predominantly composed of Tregs, marked by CTLA4 and FOXP3, whereas T-cell populations from the non-malignant sites were predominantly composed of cytotoxic T cells, expressing CCL5,

---

**Figure 1. Single cell atlas of the malignant human liver.**

A  Experimental scheme, tumor, and adjacent non-tumor liver samples from surgeries were dissociated for scRNA-seq, frozen for LCM, and fixed for smFISH.

B  tSNE plot colored by normalized sum of pan-carcinoma markers taken form Puram et al (2017). "n"—indicates the number of cells per group.

C  tSNE plot colored by the 17 Seurat clusters including hepatocytes, endothelial cells (liver sinusoidal endothelial cells—LSEC, non-tumor liver vascular endothelial cells—LVEC, and tumor liver vascular endothelial cells—LVECt), mesenchymal cells (Stellate cells, cancer-associated fibroblasts—CAFs, Pericytes, vascular smooth muscle cells—vSMC), immune cells (Kupffer cells, scar-associated macrophages—SAMs, tissue monocytes 1—TM1, cDC1, cDC2, T cells, and B cells), proliferating cells, and carcinoma cells.

D  Heatmap showing the normalized expression of marker genes for the different clusters (Materials and Methods). Expression is normalized by the maximal expression among all cell types.

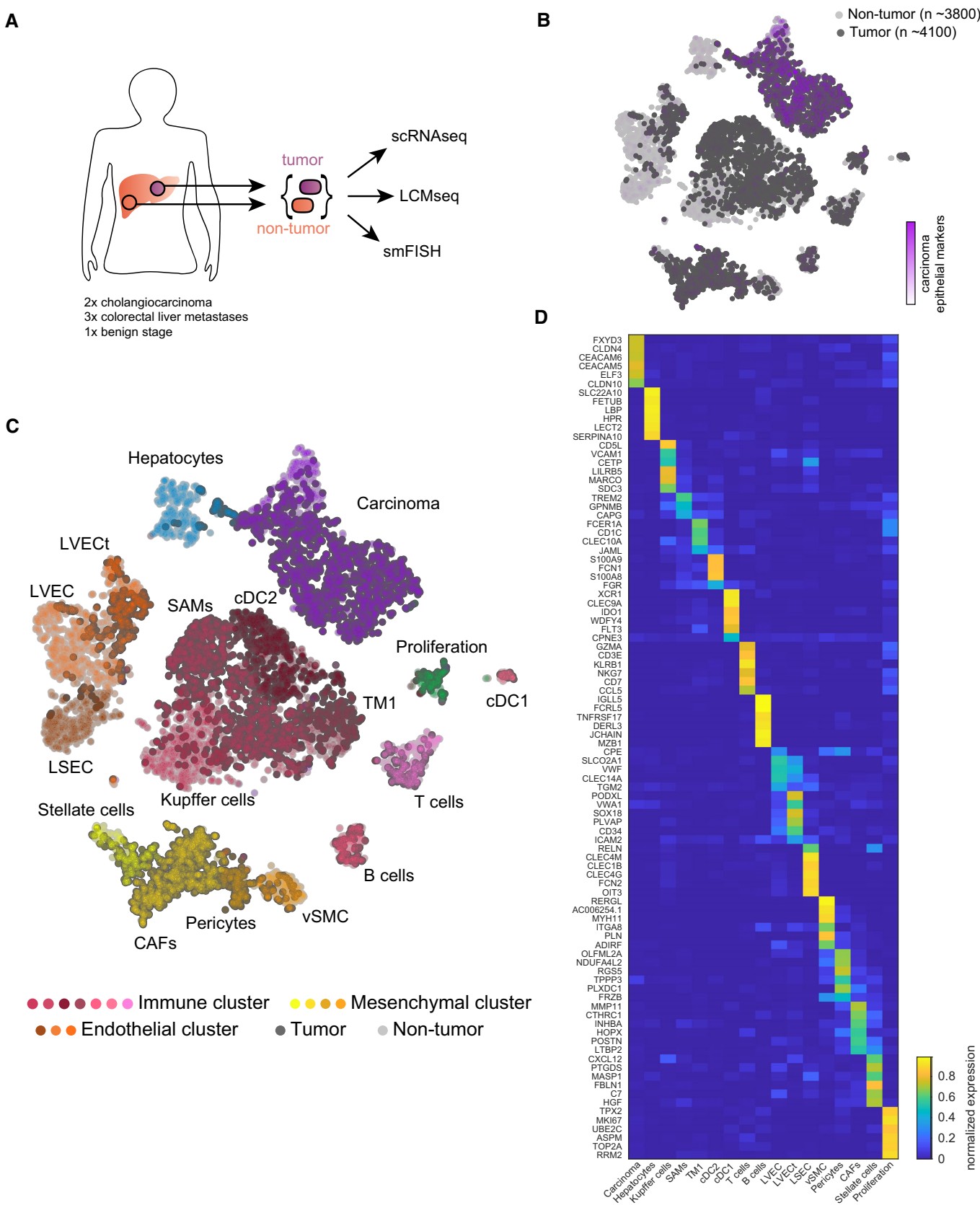

**Figure 1.**

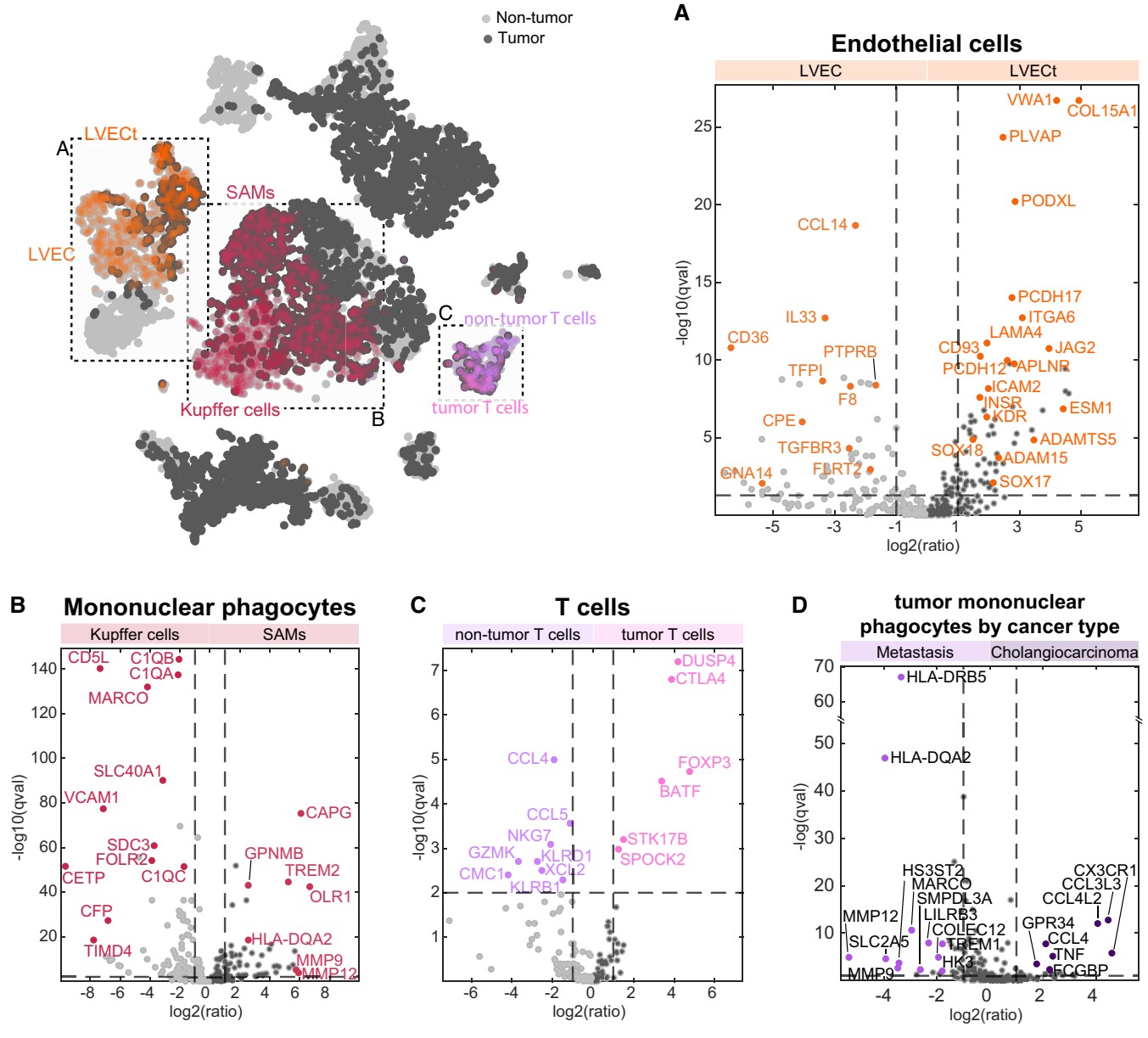

**Figure 2. Expression signatures of tumor endothelial and immune cells.**

A–D   Top-left—tSNE plot of the Seurat clusters, boxes demarcate compared clusters. Dashed boxes and labels indicate the cell clusters that are compared in panels A–C. (A) Volcano plot of differential gene expression (DGE) between liver vascular endothelial cells in the tumor and non-tumor samples. (B) Volcano plot of DGE between mononuclear phagocytes in the malignant and non-malignant samples. (C) Volcano plot of DGE between T cells in the tumor and non-tumor samples. (D) DGE analysis between tumor mononuclear phagocytes classified by cancer type (cholangiocarcinoma in dark purple and metastasis in light purple). Wilcoxon rank-sum tests were used to generate *P*-values, Benjamini–Hochberg multiple hypotheses correction was used to compute *q*-values. Labeled dots in all panels are gene names of selected differentially expressed genes between the compared two clusters.

GZMK, and NKG7 (Fig 2C). These divisions within the immune cell populations suggest a recruitment of immune-suppressive subsets of T cells and macrophages, as previously demonstrated for other tumors (Lambrechts *et al*, 2018; Binnewies *et al*, 2018). Additional immune cell types in the TME included conventional dendritic cells (cDC1 and cDC2), tissue monocytes (TM1), expressing FCN1 and S100A12 (Ramachandran *et al*, 2019), and B cells (Fig 1C, Appendix Fig S4).

We further assessed the differences in the expression signatures of endothelial cells, mononuclear phagocytes, and T cells between the tumor sites of the cholangiocarcinoma patients and the metastases patients (Fig 2D). Endothelial cells and T cells did not exhibit differential expression between these two etiologies. In contrast, mononuclear phagocytes exhibited up-regulation of chemokines such as CCL4, CCL4L2, and CCL3L3 in the cholangiocarcinoma samples and extracellular remodeling genes

such as MMP19, MMP12, and HS3ST2 in the metastatic patients.

## Diversity of the human liver mesenchymal cells

Our atlas included four mesenchymal cell clusters (Fig 3, Appendix Fig S5A). Hepatic stellate cells, marked by the retinol binding protein 1 (RBP1), and vascular smooth muscle cells, marked by Myosin-11 (MYH11) (Ramachandran *et al*, 2019), were abundant in the non-malignant liver sites (Fig 3A, Appendix Fig S1C). Mesenchymal cells in the malignant liver sites included two clusters. Cancer-associated fibroblasts (CAFs), expressing extracellular matrix (ECM) genes such as COL1A1, LUM, and BGN, formed the larger cluster. A second cluster included cells expressing classic markers of pericytes, periendothelial mesenchymal cells with important roles in regulating vascular integrity (Armulik *et al*, 2011). These markers included RGS5 and CSPG4, encoding the neuron-glial antigen 2 protein (NG2) (Armulik *et al*, 2011). We found that some previously suggested markers of pericytes, such as DES (Nehls *et al*, 1992) and ANPEP (Kumar *et al*, 2017), were not specifically expressed in pericytes in the malignant human liver context (Dataset EV3). Importantly, pericytes were almost absent from the non-malignant sites (Appendix Fig S1C). We used smFISH to demonstrate that the RGS5$^+$ cells are indeed adjacent to endothelial cells, marked by PDGFB, as expected from pericytes (Fig 3B and C). In contrast, cells expressing the CAFs marker COL1A1 resided farther away from the endothelial cells (Fig 3B and D, and Appendix Fig S5A).

Paracrine and juxtacrine interactions between endothelial cells and their attached pericytes have been shown to be important for proper vascularization (Annika *et al*, 2005). Our single cell atlas enabled unbiased identification of signaling pathways that could affect gene expression between the physically interacting endothelial cells and pericytes. To this end, we applied NicheNet (Browaeys *et al*, 2019), a computational method that predicts ligand–receptor interactions based on induction of downstream target genes (Fig 3E, Dataset EV4). We identified signaling from endothelial cells to pericytes via JAG1,2-NOTCH3, and PDGFB-PDGFRB (Fig 3E, Appendix Fig S5B), and signaling from pericytes to endothelial cells through SLIT2-ROBO1,4 (Fig 3E, Appendix Fig S5C) and ANGPT2-TEK (Fig 3E). The ligands and receptors mediating the endothelial cell–pericyte cross-talk were enriched for juxtacrine signaling pathways, angiogenesis, and chemokine and cytokine signaling (Fig 3F).

## Recurring interactions between the carcinoma cells and the tumor microenvironment

Tumor growth is highly dependent on the cross-talk between the tumor cells and the stromal cells in the TME. Stromal cells provide important growth factors and signaling molecules that enhance tumor growth and survival. In turn, the tumor cells secrete ligands sensed by the stromal cells, which facilitate their recruitment (Zhou *et al*, 2017). Our single cell atlas facilitated an analysis of the molecular cross-talk between tumor cells and each of the stromal cell types. To this end, we parsed a database of ligand–receptor interactions (Ramilowski *et al*, 2015) and identified pairs, for which the interacting proteins were specific to the carcinoma cell cluster on

the one hand and to the supporting stromal cell clusters on the other (Zhou *et al*, 2017; Halpern *et al*, 2018; Materials and Methods). We found that CAFs and SAMs were interaction hubs, representing 49.3% of all carcinoma–TME interactions (Fig 4A). We focused on specific ligand–receptor interactions that recurred in at least three of the five patients with malignant cancer (Fig 4B, Dataset EV5). The resulting tumor interactome network highlighted several recurring modules, including a large matrix remodeling module, modules centered around ERBB, HGF-MET, TGFbeta, FGF, IGF, and VEGFA, a lipid trafficking module, and a WNT planar cell polarity module (Fig 4B).

The largest module consisted of matrix remodeling proteins. The malignant ECM has a unique composition that is shaped by ECM assembly and degrading proteins, collectively known as the tumor matrisome (Naba *et al*, 2016; Varol & Sagi, 2018). Features of the ECM such as stiffness and porosity facilitate both optimal cellular contacts, maximize accessibility of growth factors and control immune cell exclusion from cancer cells (Binnewies *et al*, 2018). Within the matrix remodeling module, we found that CAFs produced most of the collagens and laminins, interacting with integrin receptors on the tumor cells (Fig 4B). Our scRNA-seq analysis further enabled identifying the secreting stromal cell type for each of the matrisome components (Naba *et al*, 2016; Appendix Fig S6).

The WNT Planar cell polarity (WNT-PCP) pathway has been suggested to promote metastases and cancer cell invasion (Wang, 2009). PCP signaling is activated by non-canonical Wnt morphogens, such as WNT5A, which we found to be expressed by both CAFs and immune cells (Fig 4B–D). CTHRC1, a secreted collagen triple helix filament that forms a complex that stabilizes WNT binding to its tumor-expressed receptor-FZD (Yamamoto *et al*, 2008), was specifically expressed by CAFs. Thus, immune cells and CAFs jointly modulate WNT-PCP tumor signaling.

We observed a similar cooperation of immune cells and CAFs within the MET signaling module. MET signaling is a major driver in hepatic tumors and metastases (De Silva *et al*, 2017). We found that HGF, the main activating ligand of MET, was expressed by both SAMs and CAFs (Fig 4B–D). DCN, encoding the decorin protein, is expressed by CAFs and in turn inhibits HGF-MET binding (Goldoni *et al*, 2009). Our interaction map further revealed an additional role of DCN as an interactor of the carcinoma-specific receptor EGFR (Fig 4B). In summary, our tumor interactome analysis revealed the details of the molecular cross-talk between the tumor and stromal cell types.

## Recurring interaction network connectivity correlates with liver tumor severity

The recurring interactions between the carcinoma cells and cells in the TME suggest that elevated expression of these ligands and their matching receptors could convey a selective advantage to cells in the liver TME. To assess this hypothesis, we examined a cohort of 383 bulk-sequenced liver tumors from the TCGA database and computed a network score based on our recurring network connectivity (Fig 4E and F). For each tumor, we first computed a score that consists of the summed products of the expression levels of each ligand and matching receptor and normalized it by computing a randomized score based on degree-preserving random networks

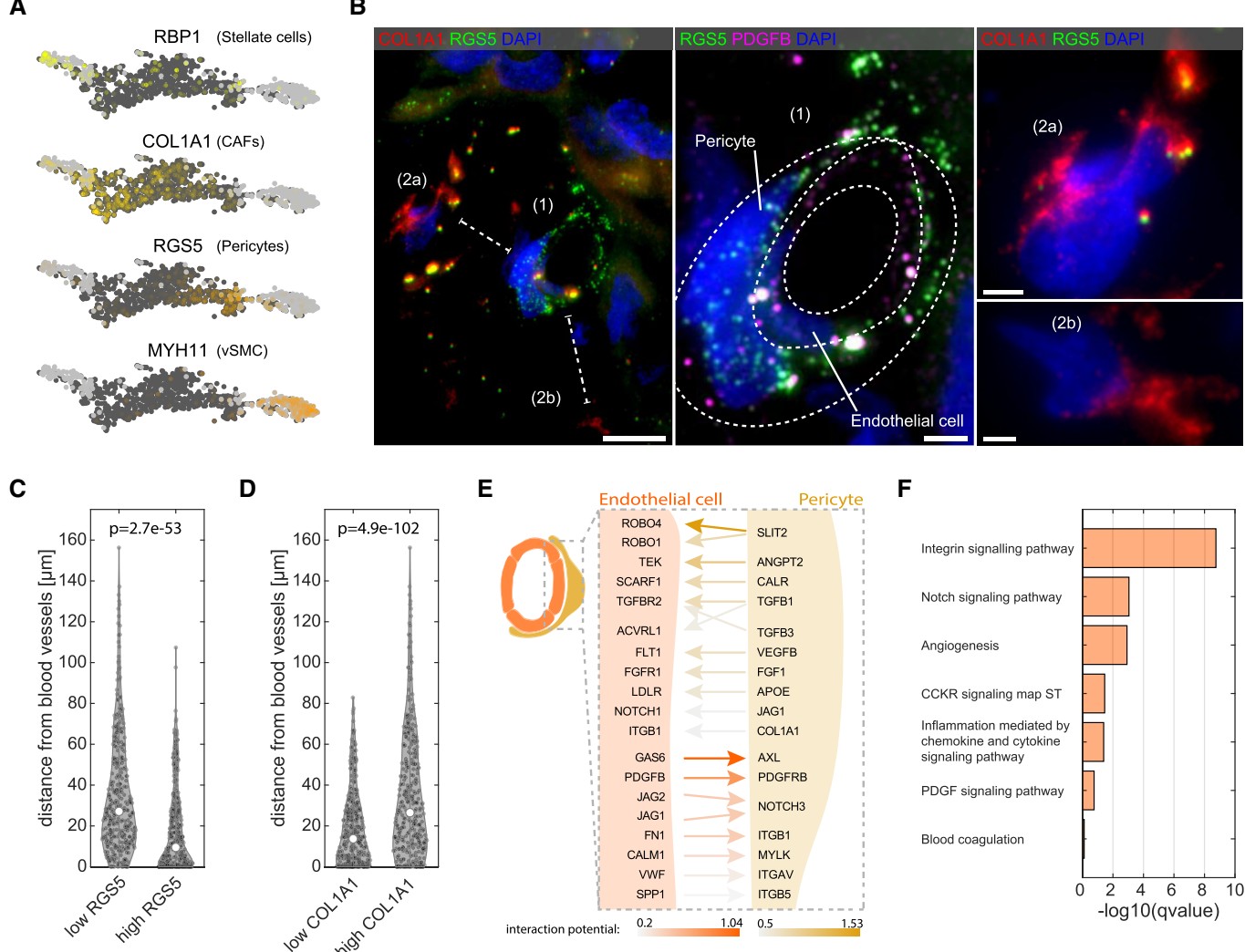

**Figure 3. Mesenchymal heterogeneity in the liver malignant sites.**

A  Key marker genes for the four mesenchymal clusters (RBP1, COL1A1, RGS5, and MYH11). Light gray dots denote cells originating from non-tumor samples. Dark gray dots denote cells originating from the tumor samples.

B  Left—Representative smFISH image of patient p1 stained for RGS5 and COL1A1 showing distinct spatial localization of CAFs and pericytes. Scale bar 10 μm. Dashed lines mark the shortest distance of cells (2a) and (2b) from the cell (1). Middle—zoom-in of (1) from left panel, showing a blood vessel like structure formed by endothelial cells marked by PDGFB (magenta) wrapped by pericytes marked by RGS5 (green). Dashed lines are two consecutive cell layers of endothelial cells and pericytes. Scale bar 2.5 μm. Right—zoom-in of (2a and b) from the left panel, showing two distant CAFs expressing high COL1A1 signal but not RGS5. DAPI used for nuclei staining. Scale bar 2.5 μm.

C, D  Violin plot of the distance from blood vessels of low/high RGS5 expressing cells ($n = 358$ and $n = 360$, respectively) and low/high COL1A1 expressing cells ($n = 359$ and $n = 359$, respectively). "p" is the P-value determined by Wilcoxon rank-sum test. Empty circles are the medians over all repeats.

E  Schematic representation of the top-ranked interaction (bona-fide) detected by NicheNet (Materials and Methods). Results are sorted by the prior interaction potential between pericytes and tumor LVECt cells.

F  Pathway enrichment analysis for all bona-fide genes (Dataset EV4) using Enrichr tool. Images in this figure are representative images out of eight independent experiments over four patients.

(Materials and Methods, Fig 4E). This normalization is important, since a high score may simply reflect elevated expression of the ligands and receptors, often oncogenes, rather than the coordinated expression of ligands and their *matching* receptors. We found that the network score significantly increased along the liver tumor stages (Fig 4F). Thus, our interaction score correlates with tumor severity.

## Spatial transcriptomics identifies zonation patterns of hepatocytes

Cells in tissues and solid tumors reside in zones that often exhibit variability in oxygen levels, nutrient availability, and morphogen concentrations. These can in turn generate spatial heterogeneity of gene expression (Moor & Itzkovitz, 2017) and

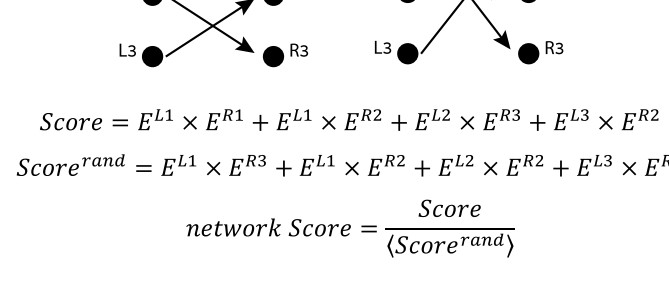

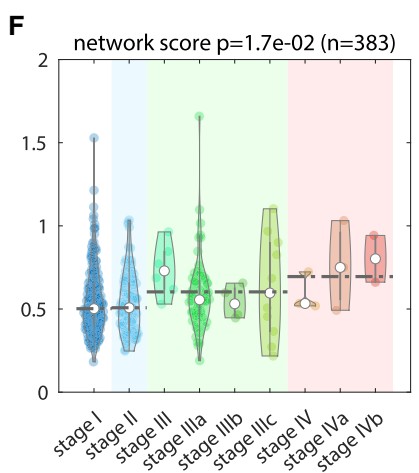

**Figure 4.   Human liver interactome delineates tumor–stroma cross-talk.**

A   Summary of the total number of ligand–receptor interactions among clusters with at least 20% tumor cells. Interactions of tumor TME cell types with the carcinoma cluster marked by red box.

B   Network of recurring ligand–receptor interactions between carcinoma cells and stromal cells from the malignant sites. Node colors denote the cell type cluster in which the ligands/receptors are enriched. Gray arrows color indicates the interaction Zscore (Materials and Methods). Recurring modules are shaded. Included are all recurring interactions that significantly appeared in at least three patients.

C   Dot-plot of selected genes highlight shared interaction motifs between different clusters colored by max normalized expression for each genes across all clusters in (A). For each gene, dot size represents the fraction of positive cells for each cluster.

D   Recurring interaction motifs between CAFs and tumor immune cells. Top—CAFs and SAMs comodulate carcinoma–stroma interaction. CAFs produce DCN that modulates the interaction between the CAF-SAMs-expressed ligand HGF and the carcinoma-expressed receptor MET. Bottom—CAFs produce CTHRC1 that modulates the interaction between the WNT5A ligand, expressed by CAFs, SAMS, and TM1 and the carcinoma-expressed receptor FZD5.

E   Strategy for computing interaction scores for each tumor. A score is computed as the sum of the products of all ligands and matching receptors of the recurring interaction network. These are compared to the average score obtained when randomizing the real interaction network in a manner that preserves the number of outgoing and incoming interactions of each ligand and receptor ($Score^{rand}$). The ratio of the real and randomized scores constitutes a network score.

F   Network score increases with increasing tumor stage. Analysis for 383 TCGA sample of liver hepatocellular carcinoma (LIHC) and cholangiocarcinoma (CHOL). *P*-value determined by Wilcoxon rank-sum test. Empty circles are the medians over all repeats. Dashed lines are the median over the tumor stage.

result in distinct spatial representation of different cell types. The liver is a spatially heterogeneous organ, composed of repeating anatomical units termed lobules, which are polarized by centripetal blood flow (Ben-Moshe & Itzkovitz, 2019). Spatially resolved single cell transcriptomics in mice revealed extensive zonation of hepatocyte gene expression along the lobule radial axis (Halpern *et al*, 2017), and pseudotime analysis of scRNA-seq data suggested that similar zonation may also be prominent in the human liver (Aizarani *et al*, 2019). We have recently developed approaches to combine scRNA-seq with transcriptomics of laser-capture microdissected (LCM) tissue to infer the zonation patterns of genes in the intestine (Moor *et al*, 2018; Halpern *et al*, 2020). We sought to apply this approach to obtain zonation patterns of human hepatocytes.

We used LCM to dissect six lobule zones within the non-malignant liver site, spanning the central vein and portal node (Fig 5A and B). We extracted RNA from each zone and performed bulk RNAseq using mcSCRBseq (Materials and Methods), a sensitive method for sequencing ultra-low mRNA levels. We used our scRNA-seq atlas to identify hepatocyte-specific genes, and among these, selected panels of pericentral and periportal landmark genes that exhibited zonated expression in the LCMseq dataset (Appendix Fig S7A, Dataset EV6, "non-malignant_liver_LCM" sheet). Using these panels, we inferred the lobule coordinates of each sequenced hepatocyte and averaged spatially resolved groups to obtain a global zonation map. Since our data contained approximately 400 hepatocytes, we applied the LCMseq reconstruction to a recently published cell atlas of the non-malignant human liver that included more than 2,500 hepatocytes (Aizarani *et al*, 2019). This enabled higher statistical power in resolving zonation patterns (Fig 5C). We validated our reconstructed zonation patterns by comparing the zonation profiles of hepatocyte-specific genes not used as landmarks to the LCMseq profiles (Appendix Fig S7B–D). Our reconstructed profiles also recapitulated zonated human hepatocyte genes previously identified using LCM (McEnerney *et al*, 2017; Appendix Fig S8A).

We found that, as observed in mice, many hepatocyte genes were significantly zonated along the lobule radial axis (2,677 genes out of 8,536 genes with expression higher than 1e-5 of cellular UMIs had *q*-value < 0.2, Fig 5C, Dataset EV7, "reconstructed_human_hepatocytes" sheet). Pericentrally zonated

processes included primary bile acid biosynthesis and metabolism of xenobiotics, whereas periportally zonated processes included oxidative phosphorylation and fructose and mannose metabolism (Fig 5D, Appendix Fig S9). We compared the zonation profiles in mouse (Halpern *et al*, 2017) and human and found both discordant and concordant profiles (Dataset EV7, "human_mouse_comparison" sheet, Fig 5D and E, Appendix Fig S9). Genes that exhibited overlapping profiles between mice and human included the pericentral genes CYP1A2 and LGR5 and the periportal genes HAL and SDS. The gene SLC2A2, encoding the main hepatic glucose transporter GLUT2, exhibited pericentral zonation, in contrast to its periportal zonation in mice. Similarly, the lipogenesis genes SREBF1, ACLY, FASN, and ACACA were pericentrally zonated in human and periportally zonated in mice (Dataset EV7, "human_mouse_comparison" sheet, Fig 5E). Additional discordant pathways included ribosomes, which were periportally zonated in human and pericentrally zonated in mouse, and complement and coagulation cascades, which were periportally zonated in mouse and pericentrally zonated in human (Fig 5E, Appendix Fig S9). Our map further revealed zonated transcription factors such as the pericentral ZNF101, AHR, and TBX3 and the periportal ID1, TBX15, and SOX4, as well as zonated surface markers (Appendix Fig S8B and C). The high-resolution hepatocyte zonation map forms a resource for exploring spatial heterogeneity in the human liver.

## Spatial distributions of TME populations

Solid tumors have been shown to exhibit variability in cell composition and expression programs as a function of the location within the tumor (Giesen *et al*, 2014; Angelo *et al*, 2014; Keren *et al*, 2018; preprint: Moncada *et al*, 2018). This heterogeneity is often dictated by cell proximity to spatial landmarks that include the tumor boundary, blood vessels (Kumar *et al*, 2019), and immune cell aggregates (Colbeck *et al*, 2017). We next asked whether different TME cell types in our atlas are more abundant in distinct tumor microenvironments.

We used LCM to dissect 63 tissue regions from four patients. These regions included the tumor border (tb), tumor core (tc), the border between the tumor and the fibrotic regions (ftb), tumor islets within the fibrotic regions (ti), and fibrotic zones (fz, Fig 6A and B).

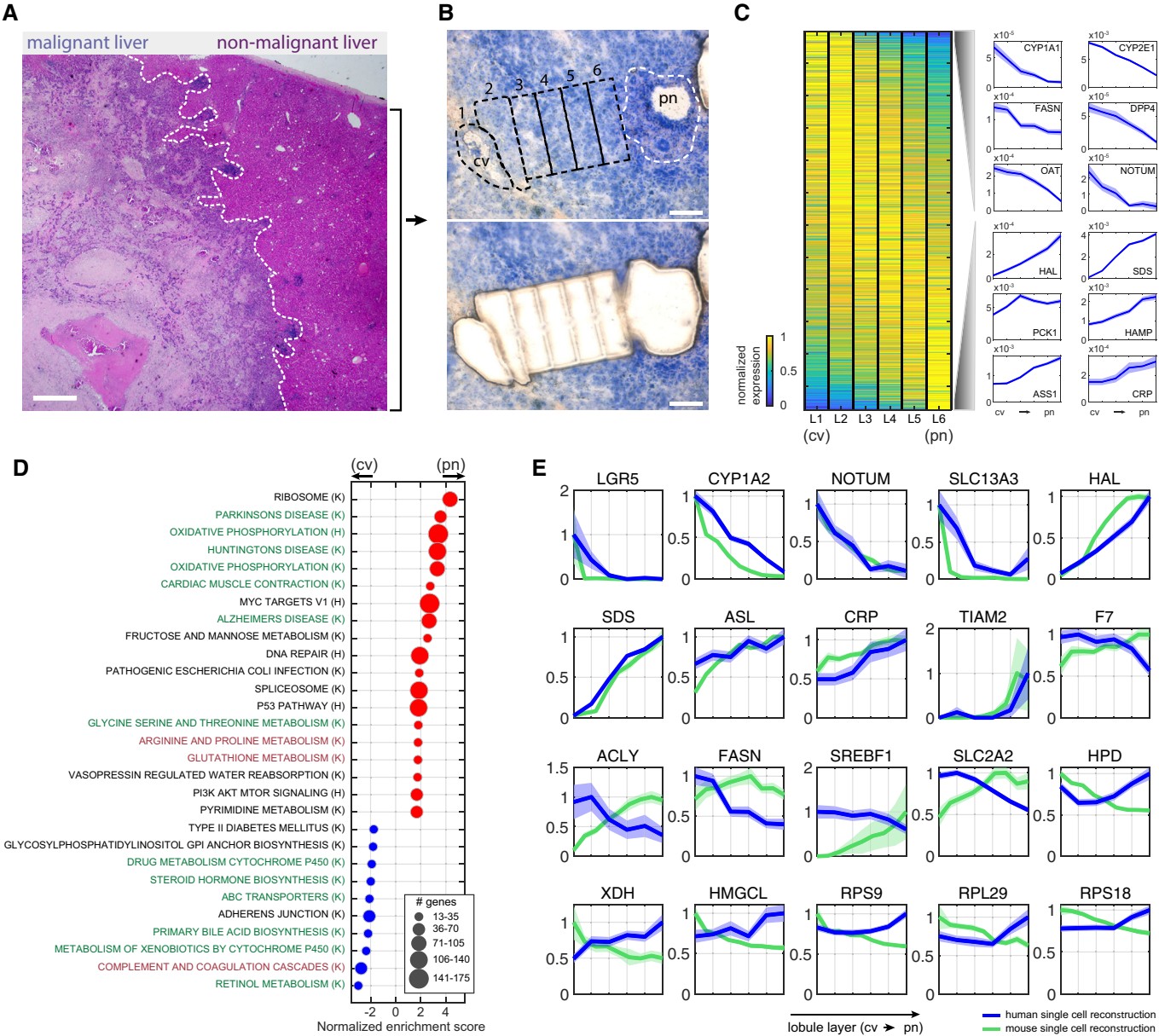

**Figure 5. Spatial analysis of the malignant and non-malignant human liver.**

A  H&E staining of malignant and non-malignant sites from patient p4, separated by the white dashed line. Scale bar is 100 μm.

B  Non-malignant human liver lobule from patient p4 used for LCM stained with HistoGene Staining Solution (Materials and Methods). Top—sequential zones from the central vein (cv) to the portal node (pn) before microdissection. LCM captured zones marked by the black dashed line. White dashed line marks the boundaries of NPC (non-parenchymal cells) located in the pn and not included in the analysis. Bottom—Tissue after microdissection. Scale bar is 100 μm.

C  Zonation of human hepatocytes. Left—Max normalized expression over the reconstructed six lobule layers (L1–L6), sorted by center of mass from the central vein (cv) to the portal node (pn). Right—selected pericentrally (top), and periportally (bottom) zonated genes. Blue patches are standard errors of the mean.

D  Gene set enrichment analysis of genes with maximal expression above 1e-5 normalized UMI counts. Centrally (cv) enriched sets are marked with blue dots, and portally (pn) enriched sets are marked with red dots. Green fonts denote pathways that are concordantly zonated in mouse, black fonts denote pathways that are not zonated in mouse, and red fonts denote pathways that are inversely zonated in mouse. "K" denotes gene sets obtained from KEGG database, and "H" denotes gene sets obtained from Hallmarks database.

E  Zonation profiles of selected genes from human (blue) and mouse (green). Each profile is normalized by the maximal expression across zones. Patches are standard errors of the mean.

Data information: Mouse data used in (D and E) were obtained from Halpern *et al* (2017).

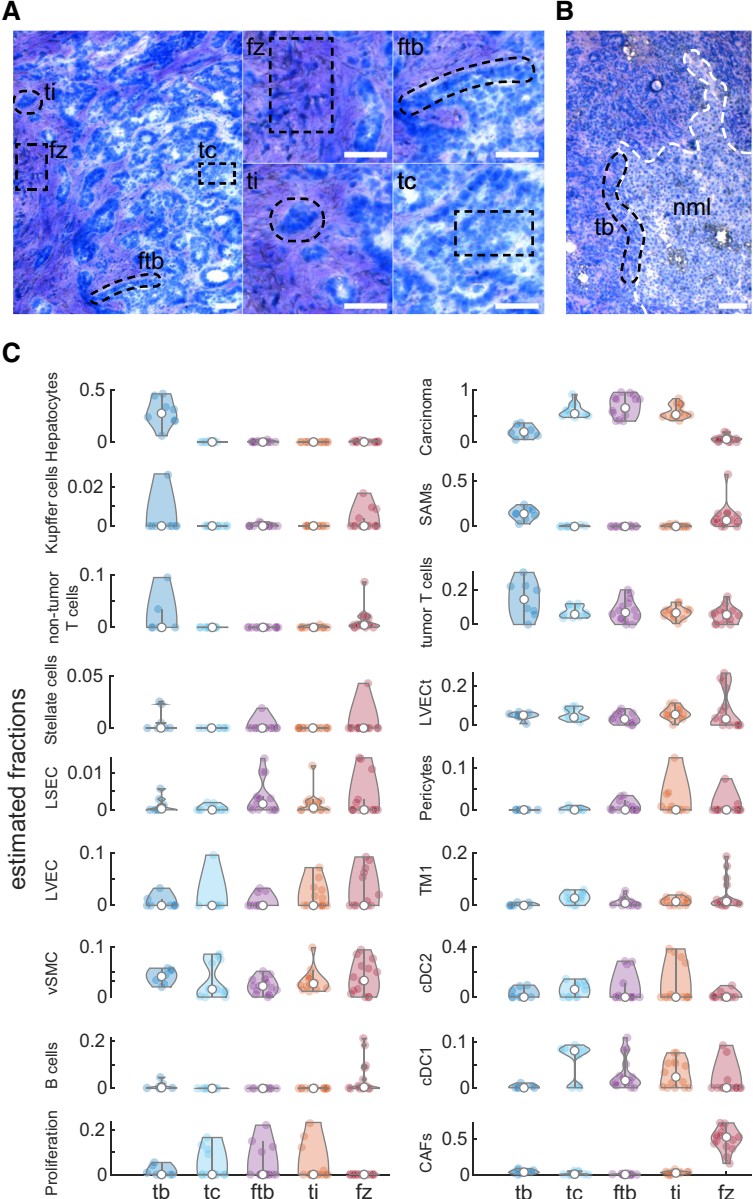

**Figure 6. Spatial distribution of the TME populations.**

A, B   Tissue sections from patients p1 and p3 stained with HistoGene Staining Solution. Laser-captured zones are marked with dashed black line including: tumor core (tc), the border between the tumor and the fibrotic regions (ftb), tumor islets within the fibrotic regions (ti) and fibrotic zones (fz) in panel (A), and tumor border (tb) in panel (B). Small images in (A) are magnifications of the corresponding zones in the large image for the left. White dashed line in (B) marking the non-malignant liver (nml) border. Scale bars in all panels are 100 μm.

C   Spatial distribution of the different TME populations across the analyzed five zones. Values are the estimated proportions based on deconvolution of the transcriptome of each zone with the average transcriptome of the cell types (Materials and Methods). Y-axes show the estimated fractions of the cells, and empty circles are the medians over all repeats (n = 8 for tb, 11 for tc, 15 for ftb, 14 for ti and 15 for fz).

We performed bulk RNAseq on these regions (Dataset EV6 "malignant liver LCM" sheet) and used the expression signatures of the cell types in our single cell atlas to estimate the proportions of each cell type, based on the bulk measurements. To this end, we used AutoGeneS (preprint: Aliee & Theis, 2020) to deconvolve the data (Fig 6C, Dataset EV9, Materials and Methods). We found that different cell types differentially populated different tumor zones. As

expected, hepatocytes were most abundant in the tumor border zones and almost completely depleted from other tumor zones. CAFs were most abundant in the fibrotic zones. Pericytes showed spatial abundances that highly overlapped those of LVECt, as expected based on their physical attachments (Fig 3). Notably, SAMs and both tumor T cells and non-tumor T cells showed higher abundance in the tumor border zones (Fig 6C).

# Discussion

Our study combined single cell transcriptomics with spatial analysis techniques to reconstruct a cell atlas of the malignant and non-malignant human liver. Cancer treatment has been hampered by tumor heterogeneity, attributed to patient-specific somatic mutations and epigenetic changes. This heterogeneity raises a challenge of tailoring personalized treatments to match the unique properties of each patients' tumors (Bedard *et al*, 2013). Our analysis of the cross-talk between the TME and carcinomas revealed recurring interaction modules between different patients. This finding suggests that drugs targeting stromal cells or their mechanisms of cross-talk with the tumor could potentially be more broadly applicable. Our ligand–receptor interaction maps (Fig 4, Dataset EV5) provide the tools to identify potential targetable candidates that could perturb tumor–stroma cross-talk.

Tumor fitness depends on proper vascularization, which in turn is strongly affected by the proper attachment and function of pericytes and endothelial cells (De Palma *et al*, 2017). Our study identified the molecular interactions between endothelial cells and pericytes in the malignant human liver. SLIT-ROBO signaling was previously shown to be important for endothelial–pericyte attraction in the formation of blood vessels in malignant human cell lines (Wang *et al*, 2003). Our analysis highlighted a potential role of this pathway, composed of the pericyte SLIT2 ligand and endothelial ROBO receptor in maintaining endothelial–pericyte interaction in the TME. Additional pathways included delta-notch and VEGF signaling, previously shown to be important blood vessels formation (Pitulescu *et al*, 2017), and PDGFB-PDGFRB signaling, shown to facilitate pericyte recruitment (Raza *et al*, 2010). This analysis of pericyte–endothelial cross-talk could form a basis for potential interventions aimed at perturbing vascular integrity by targeting endothelial–pericyte cross-talk.

Diverse methods for reconstructing spatial gene expression profiles in tissues have been developed in recent years (Moor & Itzkovitz, 2017). The combination of single cell RNAseq with external spatial measurements of landmark genes using technologies such as LCMseq (Moor *et al*, 2018) or smFISH (Halpern *et al*, 2017) has proven to be particularly accurate. Our measurements of transcriptomes of laser-capture microdissected regions along the liver lobule pericentral axis enabled reconstructing a zonation map of human hepatocytes with high spatial resolution. This zonation analysis could assist in modeling metabolic function in the human liver (Gille *et al*, 2010). Moreover, zonated surface markers that we have identified (Appendix Fig S8C) could be used for bulk measurements of spatially stratified human hepatocyte populations, to obtain the zonated features of proteins, metabolites and other cellular properties (Ben-Moshe *et al*, 2019). Our spatial transcriptomics of the malignant site demonstrated higher abundance of immune cell types, specifically T cells and SAMs, in the tumor border (Fig 6C). This localization pattern resembles structured immune microenvironments previously observed in breast tumors (Keren *et al*, 2018).

Recent work has begun to lay the ground for a comprehensive human cell atlas, consisting of a blueprint of all cell types in the human body (Regev *et al*, 2017). Similar single cell atlases in human tumors are essential for understanding the contribution of different cells to the malignant process (Tirosh *et al*, 2016; Puram *et al*, 2017; Zheng *et al*, 2017; Lambrechts *et al*, 2018; Azizi *et al*, 2018). Our study constitutes a step in this direction, by characterizing the cell types in the non-malignant and malignant human liver.

# Materials and Methods

### Patients

This study was approved by the institutional review board committees of the Weizmann Institute of Science and Hadassah hospital, Israel (0327-17-HMO). Informed consent was obtained from all samples donors included in the study. Patients with colorectal metastasis, cholangiocarcinoma, or liver cyst were included in the study. All samples were collected from isolated tumors immediately after surgical isolation (Dataset EV1). The fibrotic status of the non-malignant livers was assessed by a pathologist, using Masson Trichrome stain (Dataset EV1). The only patient that exhibited substantial fibrosis was the cholangiocarcinoma patient p2.

### Sample collection

Samples were collected from the surgery room immediately after isolation into sterile saline solution and processed in a sterile cabinet. In each surgery, two distinct samples were obtained—one from the tumor region ("malignant site") and another from an adjacent non-tumor region that was at least 1-5cm from the tumor region ("non-malignant site"). Each sample was divided equally into three pieces as follows: (i) sample for scRNA-seq stored in cold 1× DMEM, supplemented with glutamine (GIBCO, 35050-038), (ii) sample embedded into OCT (Scigen, 4586) block and kept on dry ice, and (iiii) sample fixed in cold 4% paraformaldehyde (PFA, Santa Cruz Biotechnology, sc-281692) in PBS on ice for 3 h (Fig 1A).

### Tissue fixation and smFISH

Tissue was fixed in 4% PFA for 3 h at 4°C and then transferred to 30% sucrose solution in 4% PFA for overnight at 4°C. Fixed tissues were embedded in OCT and stored at −80°C for later use. Five μm sections of fixed human tissue were mounted onto poly L-lysine coated coverslips and used for smFISH staining. Probe libraries were designed using the Stellaris FISH Probe Designer Software (Biosearch Technologies, Petaluma, CA), see Dataset EV10. smFISH was performed according to a previously published protocol (Itzkovitz *et al*, 2011) with the following modification: Sections were digested with proteinase K at 50°C for 10 min for better cellular permeability before the wash buffer denaturation steps. DAPI (Sigma-Aldrich, D9542) was used for nuclear staining. smFISH imaging was performed on a Nikon-Ti-E inverted fluorescence microscope equipped with 100× oil immersion objective and a Photometrics PRIME 95B 25 mm sCMOS camera. Probe libraries for messenger RNAs of interest were coupled to Cy5, Alexa594, or TMR. To stain carcinoma cells, tissues were incubated with pan-cytokeratin antibody (1:100, Invitrogen, 53-9003-82) conjugated to Alexa Fluor 488 in GLOX buffer for 20 min at room temperature and washed once by fresh GLOX buffer before mounting.

## Single cell preparations and isolation

Isolated human liver samples were chopped into small parts using sharp scalpel in fresh 1× DMEM supplemented with glutamine (GIBCO, 35050-038). All parts were placed in preheated enzymatic solution and digested mechanically according Liver Dissociation Kit (GentelMAX, 130-105-807) with small modifications. Liberase Blendzyme3 recombinant collagenase (Roche Diagnostics) were added to the enzymatic cocktail and preheated in 37°C for 10 min. After adding the liver parts, mechanical digestion was performed using m_liver_03 built-in program as suggested by the protocol using GentelMAX Dissociator (GentelMAX, 130-093-235) and incubated for 30 min at 37 C with gentle mixing using a rotator. An additional mechanical digestion step was applied after the incubation using m_liver_04 built-in program, followed by filtering the mixture using 100 μm nylon mesh (FALCON, 352360) as mentioned in the protocol. The cells were concentrated by centrifugation for 10 min at 100 $g$ and resuspended in cold FACS buffer (2 mM EDTA pH 8, 0.5% BSA in 1× PBS). The concentrated cell suspension was taken directly for sorting. Forward scatter (FSC) and side scatter (SSC) were calibrated to exclude debris. Dead cells were excluded using PI staining (1:1,000 $v/v$, Thermo Fisher P3566). Single cells were collected directly to MARS-seq lysis solution (Jaitin et al, 2014) in a format of 384-well plate and stored at −80°C (see MARS-seq library preparation).

## MARS-seq library preparation

Single cell libraries were prepared, as described previously (Jaitin et al, 2014). Briefly, mRNA from cells sorted into MARS-seq capture plates was barcoded and converted into cDNA and pooled using an automated pipeline. The pooled sample was then linearly amplified by T7 in vitro transcription, and the resulting RNA was fragmented and converted into sequencing ready libraries by tagging the samples with pool barcodes and Illumina sequences during ligation, reverse transcription, and PCR. Each pool of cells was tested for library quality, and concentration was assessed as described in Jaitin et al (2014). Machine raw files were converted to fastaq files using bcl2fastq package, and to obtain the UMI counts, reads were aligned to the human reference genome (GRCh38.91) using zUMI package (Parekh et al, 2018) with the following flags that fit the barcode length and the library strandedness: -c 1-7, -m 8-15, -l 66, -B 1, -s 1, -p 16. Downstream analysis done as explained in "scRNA-seq analysis" part.

## scRNA-seq analysis

scRNA-seq analysis of the unique molecular identifier (UMI) counts for exon mapped reads produced by the zUMI pipeline was processed with Seurat 3.1 (Satija et al, 2015) running in R3.5.1 for each patient individually. Cells with UMI counts below 200 or higher than 3,000 or mitochondrial content above 35% were removed. This analysis resulted in 7,947 cells (with median of 797 UMIs and 430 detected genes per cell). The data were log-normalized according to default Seurat settings. Variable genes for principal component analysis were identified using the Seurat function "FindVariableFeatures" using the "vst" method with the default parameters. UMI number and mitochondrial gene content were regressed out.

To combine the Seurat output for all patients, integrated data object were generated using single cell integration analysis (Stuart et al, 2019). Integration anchors and combined data were calculated with "dims = 50" parameter. Principle component analysis was performed on the expression levels of the detected variable genes. The first 30 principal components were included for further downstream analyses based on Seurat's "JackStrawPlot" function. T-distributed stochastic neighbor embedding (tSNE) was used to visualize the computed clusters. Clustering was performed with Seurat "FindClusters" function, using the first 30 principle components and resolution of 1.

Annotation for detected clusters was based on highly expressed genes for each cluster and performed in two stages. In the first stage, markers were detected by comparing each cluster to all other clusters in the atlas (Dataset EV2, general sheet). This resulted in some markers that were found in several clusters (e.g., COL1A1 that marks CAFs, stellate cells, and pericytes). To further refine the cluster annotation, we grouped the 17 Seurat clusters into coarse-grained clusters based on the known pan-markers of immune cells (PTPRC), endothelial cells (PTPRB), and mesenchymal cells (TAGLN). We then identified coarse-grained marked by comparing each cluster to the other clusters within its coarse-grained group (Dataset EV2, type-specific markers sheet). Carcinoma clusters were merged for downstream analysis. The list of cell type markers (Dataset EV2) was obtained by running the Seurat function "FindAllMarkers" with the following parameters: log-fold change above 0.25, minimum fraction of cells expressing a gene in either of the compared populations is 10%, minimum cells expressing a gene is 10, positive genes only. The concise list of cell type-specific markers (Fig 1D) was obtained as follows: We performed Wilcoxon rank-sum tests between each cell type cluster and all other clusters and retained genes that had a minimal expression of 5e-5 of cellular UMIs in the respective cluster, expressed in more than 20% of the cluster cells and at 4-fold higher expression compared to the other clusters and $q$-value less than 0.01. Next, we generated the following coarse-grained clusters: hepatocytes, fibroblasts, immune cells, cancer cells, endothelial cells, and proliferating cells. We retested the retained marker genes against other cell type clusters from the same coarse-grained cluster and retained genes with 10-fold higher expression compared to the mean expression in the other cell types within the respective coarse-grained cluster. The top marker genes of each cluster are presented in Fig 1D.

## Cluster stability

Expression signatures for each cell type were calculated for all possible combinations of patients and were compared to the signatures based on the full atlas using Spearman correlations. Only genes with expression above 5e-6 were included in the correlation analysis. To examine the gain in correlation when adding new patients, we constructed two subsampled datasets: (1) datasets that include all possible combinations of $n < 6$ patients, where $n = 1, 2, 3, 4, 5$. We additionally included three bootstrap iterations for each patient combination. Values for the six-patient group were obtained by bootstrapping all cells. (2) For each set in (1) equally sized subsets of sampled cells from the complete atlas, while ignoring the patient

information. Set (2) served as a control to assess the decrease in gene expression signature correlation that arose from the decrease in the number of cells, rather than patients. The two calculated correlations converged, with an exception of the carcinoma cluster, where the gene expression signatures changed with each new added patient. In this analysis, T cells were split into two cell populations according to their sample origin (tumor and non-tumor), as were Cholangiocytes, which were separated from the carcinoma cells according to their non-tumor origin.

## Quantification of RGS5 and COL1A1 from smFISH images

Two sets of cells were selected per image for this analysis: (i) all PDGFB expressing cells and (ii) randomly selected COL1A1/RGS5 expressing cells (30 cells on average per image) to reach 718-segmented cells from eight independent images of three different patients—p1, p2, and p4. For COL1A1/RGS5 expressing cells, we calculated the intensity of COL1A1 signal, cell area, the number of RGS5 dots, and the distance from the closest PDGFB expressing cell. To subtract the background signal from COL1A1 intensity value, the mean intensity of the bounding rectangle without cell area was used as a local background intensity for each cell. RGS5 expression was computed as the number of smFISH dots divided by the cell volume (dot concentration). COL1A1 corrected intensity values and RGS5 dot concentrations were split into two groups: lower 50 percentile and the upper 50 percentile and designated as low/high COL1A1/RGS5, respectively.

## NicheNet interaction analysis and pathway enrichment

Malignant cells of the combined Seurat cluster were tested for potential interaction between pericytes and LVECm clusters using NicheNet (Browaeys *et al*, 2019). The pipeline was run as detailed by the authors with minor modifications. For LVECm as sender cells and pericytes as receiver cells, we included genes with mean expression above 1e-5 of cellular UMIs and present in more than 10% of the cells in either cluster. For pericytes as sender cells and LVECm as receiver cells, we included genes with mean expression above 1e-5 of cellular UMIs and present in more than 5% of the cells in either cluster. This lower threshold was used to increase the number of background genes due to the smaller number of the malignant LVECm cells compare to pericytes. Genes of interest to be potentially affected by cell–cell interaction in both comparisons were selected as follows. First, we performed Wilcoxon rank-sum test between each cluster and all other clusters. We retained genes that had a minimal expression of 1e-5 of cellular UMIs in the respective cluster and had an expression of 2-fold higher compared to the other clusters and *q*-value less than 0.01. We grouped each cluster to one of the following parent clusters: hepatocytes, fibroblasts, immune cells, cancer cells, endothelial cells, and proliferating cells. Next, we retested the retained genes against other malignant mesenchymal cells in the case of pericytes (resulting in 40 specific genes) or malignant endothelial cells in the case of LVECm (resulting in 31 specific genes). Wilcoxon rank-sum test was used in all comparisons. Further steps in the pipeline kept with no changes. All predicted ligands of the ligand activity analysis were used for the downstream analysis. The first 600 target genes were used to score ligand activity. Enrichr (Chen *et al*, 2013; Kuleshov *et al*, 2016) was used to

identify enriched pathways among the ligands and receptors, using the pooled "bona-fide" set of ligands and receptors. Results in Fig 3F are taken from a metabolic, cell signaling, and pathway database Panther 2016 (Mi *et al*, 2019).

## Matrisome analysis

Marker genes of stromal clusters with malignant cells above 20% were extracted from the single cell data. Marker genes were selected to have a mean expression above 1e-5 of cellular UMIs and three times higher than the maximal mean expression in the other clusters. Next, we looked for matrisome genes that were included in the generated maker genes list (extracted from Naba *et al*, 2016, 1,027 matrisome-related genes). For the purpose of this analysis, we merged stellate malignant cells with CAFs malignant cells. Proliferation cluster was excluded from the analysis. We performed hierarchical clustering on the retained genes using the clustergram function in MATLAB version 2019b, computed over a Zscore matrix of the sum-normalized UMI count table using default parameters. Genes were sorted by the mean expression for the respective matrisome subclass (Appendix Fig S6A). Visualization of the clustering shown in Appendix Fig S6B used only genes with Zscore > 1 or Zscore < -1.

## Ligand–receptor analysis

Ligand–receptor analysis was performed similar to Bahar Halpern *et al* (2018). Briefly, a list of ligand–receptor pairs was extracted from Ramilowski *et al* (2015) (708 unique ligands and 691 unique receptors). We calculated the average of the logarithm of the UMI-summed normalized expression, $x_g^c$ for each gene g in each cluster $c$ over all cells derived from both the malignant and the non-malignant tissues. Clusters with less than 15 cells were filtered out. We computed a Zscore, $Z_g^c$, representing the enrichment of each ligand and receptor in each cluster:

$$Z_g^c = \frac{x_g^c - \text{mean}(x_g^c)}{\text{std}(x_g^c)} \tag{1}$$

where the means and standard deviations were computed over all cluster cells. We next defined an interaction score as:

$$Z_{\text{interaction}} = \sqrt{\left(Z_L^{c1}\right)^2 + \left(Z_R^{c2}\right)^2} \tag{2}$$

where $Z_L^{c1}$ is the ligand Zscore for cluster $c1$, and $Z_R^{c2}$ is the receptor Zscore for cluster $c2$. The resulting list of interactions was filtered using the following parameters: The minimum number of cells expressing either the ligand or the receptor was above seven cells, the average expression of either the ligand or the receptor was higher than 1e-5 of the cellular UMIs, $Z_L^{c1}$ and $Z_R^{c2}$ were positive and $Z_{\text{interaction}}$ was above 1.5. For the purpose of this analysis, the carcinoma clusters were combined and defined as a single tumor cluster, and the malignant stellate cells were combined with the malignant CAF cells. To focus on interactions that occur in the malignant site, we considered clusters with more than 20% malignant cells. Next, we focused on recurring interactions between carcinoma cells and stromal cells in different patients (Dataset EV5,

"carcinoma_stroma_interactions" sheet, 1,709 interactions). Patient with benign stage, p6, was excluded as it had no malignant cells (Dataset EV1). We counted the number of unique interactions between carcinoma cells and non-carcinoma cell types in different patients, and interactions that appeared in three or more patients were retained (Dataset EV5, "recurring_interactions" sheet, 335 interactions). The highest Zscore for the recurring interactions was assigned in the unique list of recurring interactions (Dataset EV5, "unique_recurring_interactions" sheet, 101 interactions). Proliferation cluster was excluded from the analysis. Cytoscape (Shannon *et al*, 2003) was used to visualize the "unique_recurring_interactions" list.

### Network score

A score was computed based on the recurring interactions (Fig 4B) as the sum of the products of the expression of all ligands and matching receptors. Additionally, a randomized score $Score^{rand}$ was computed as the mean Score obtained from 1,000 degree-preserving randomized networks (preserving the number of outgoing and incoming interactions for each ligand and receptor). The randomized networks were generated using mfinder (v1.2) (Milo, 2002) with the command "mfinder -r 1000 -ornet -nsr 10". The network score was defined as the ratio between Score and $Score^{rand}$ (Fig 4E). We computed this score on bulk RNAseq samples of primary liver tumors downloaded from the TCGA database. Tumors used included untreated individuals or individuals not reported as treated (383 samples). Kruskal–Wallis test was conducted to assess the significance of the network score between different tumor stages—tumor stage i, ii, iii (grouped with tumor iii substages), and iv (grouped with tumor substage) (Fig 4E).

### Laser-capture microdissection

Tissue blocks for laser-capture microdissection were briefly washed in cold PBS and embedded in OCT on dry ice. LCM protocol was applied as previously described by Moor *et al* (2018) with minor modifications. Briefly, 12 μm thick sections were cut from the frozen block, mounted on polyethylene-naphthalene membrane-coated glass slides (Zeiss, 415190-9081-000), air-dried for 1min at room temperature, washed in 70% ethanol for 30s, incubated in water for 30 s (Sigma-Aldrich, W4502), stained with HistoGene Staining Solution for 100 s (Thermo Fisher Scientific, KIT0401), and washed again in water for a of 30 s. The stained sections were dehydrated with subsequent 30-s incubations in 70, 95, and 100% EtOH and air-dried for 90 s before microdissection. Tissue sections were microdissected on a UV laser-based PALM-Microbeam (Zeiss). To ensure minimal damage to the surrounding cells, laser intensity and focus were calibrated before each session using Zeiss calibration wizard supplemented with the LCM operating software (Zeiss).

Manual detection of analyzed regions in each tested slide and labeling of the desired areas was done with PALM 10× and 20× lenses. Tissue fragments were catapulted and collected in 0.2-ml adhesive cap tubes (Zeiss, 415190-9191-000) containing 7 μl of lysis buffer (RLT buffer (QIAGEN, 79216) with 1% 2-Mercaptoethanol). Parameters for this step were calibrated by the automatic program wizard. Each capture section was visually confirmed by focusing the PALM on the targeted adhesive cap after the collection session and immediately stored at −20°C. Since the non-malignant liver

samples exhibited profound immune cell infiltration in the most periportal zone, we dissected six zones from the non-malignant liver (nml) forming consecutive sections of the hepatic lobule from the central vein to the periportal immune layer (Fig 5B). In the malignant sites, 63 samples were dissected from five zones from patients p1, p2, p3, and p4. These zones included the following: tumor border (tb), tumor core (tc), the border between the tumor and the fibrotic regions (ftb), tumor islets within the fibrotic regions (ti), and fibrotic zones (fz, Fig 6A). A total area of 50,000–70,000 μm² was collected per zone, 3–4 replicates.

### LCMseq for bulk LCM samples

Tissue samples collected from the adhesive caps, dissolved, and mixed with additional 7 μl lysis buffer (RLT buffer (QIAGEN, 79216) with 1% 2-Mercaptoethanol). Lysate was washed using 7 μl AMPure XP bead (BECKMAN COULTER, A63881). RNA libraries from the bulk tissues were prepared using mcSCRBseq protocol (Bagnoli *et al*, 2018) with minor modifications. RT reaction was applied directly on the beads with a final volume of 10 μl. 4.2 μl of Rnase free water was added to the beads and mixed with 4.8 μl reaction buffer (1× Maxima H Buffer, 1 mM dNTPs, 2 μM TSO* E5V6NEXT, 7.5% PEG8000, 20 U Maxima H enzyme, 1 μl barcoded RT primer). Subsequent steps were applied as mentioned in the protocol. Library final concentration of 2 pM was loaded on NextSeq 550 (Illumina) sequencing machine aiming for 20 M reads per sample. Raw files were converted to FASTQ files using bcl2fastq package, and to obtain the UMI counts, fastq reads were aligned to the human reference genome (GRCh38.91) using zUMI package (Parekh *et al*, 2018) with the following parameters: RD1 16bp, RD2 66bp with a barcode (i7) length of 8 bp. Mitochondrial genes and hemoglobin genes were removed. Fractions by UMI counts were obtained by normalizing by the sum of the UMIs per samples. Genes with normalized expression below 1% of the total UMIs were used for this normalization step.

### Hepatocyte zonation reconstruction

We used our single cell atlas to identify hepatocyte-specific genes (genes with maximum normalized UMI count above 1e-5 and 2-fold higher mean expression in the hepatocyte cluster compared to the maximum mean expression of the non-malignant cells in other clusters). This step resulted in 414 hepatocyte-specific genes. Next, we calculated the center of mass (weighted average) for the hepatocyte-specific genes based on the LCMseq data. Genes with maximum expression above 1e-5 were retained. We divided the retained genes into two groups: pericentral landmarks—genes with center of mass below 2.5 (13 genes, denoted {$pc$}) and periportal landmarks—genes with center of mass above 4.5 (13 genes, denoted {$pp$}, Appendix Fig S7A). We next used the expression of these landmark genes to infer the lobule zones of single sequenced hepatocytes from Aizarani *et al* (2019). To this end, we computed a unit-less coordinate η value for the each cell using the following equation:

$$\eta^i = \frac{\sum\limits_{j \in \{pp\}} E_j^i}{\sum\limits_{j \in \{pp\}} E_j^i + \sum\limits_{j \in \{pc\}} E_j^i} \tag{3}$$

where $E_j^i$ is the UMI-sum-normalized expression of gene $j$ in cell $i$. The η values were divided into six equal bins, representing lobule zones, and gene expression was averaged over all cells in each bin to obtain zonation values. Significantly zonated genes were assessed using Wilcoxon rank-sum test. Benjamini–Hochberg false discovery rate was used to obtain $q$-values for genes with maximal zonation value above 1e-5 of the reconstructed mean expression over the six zones (Fig 5C).

To assess the accuracy of our reconstruction, we compared the centers of mass of our reconstructed zonation profiles to the ones measured by laser-capture microdissection. To this end, we considered all significantly zonated hepatocyte genes but excluded the landmark genes. We calculated the Spearman correlation between the two obtained center of mass datasets (Appendix Fig S7B). Correlation coefficient was tested against Spearman Correlation coefficients calculated by randomized assignment of the center of mass between the single cell and laser-capture microdissection datasets (Appendix Fig S7C). $P$-values were computed using the normal distribution for the Zscores ((real correlation − mean randomized correlation)/std(randomized correlations)). In addition, we visualized the expression of genes previously shown to be differentially expressed in LCMseq of portal and central human liver zones (McEnerney *et al*, 2017) (Appendix Fig S8A).

**Hepatocyte pathway enrichment analysis and mouse comparison**

Hepatocyte-expressed genes with a maximal expression level higher than 1e-5 of sum-normalized UMI counts sorted by the center of mass were included as input to gene set enrichment analysis (GSEA) program to identify enriched pathways against Kegg and Hallmark pathways (Fig 5D). Genes of the enriched pathways are listed in Dataset EV8. Annotations of the pathways according to their zonation status in mice were performed using Dataset EV5 in (Halpern *et al*, 2017). Comparison of the mean zonation profiles of KEGG pathways were performed as in (Halpern *et al*, 2017). In short, hypergeometric $P$-value of 186 Kegg pathways was calculated for expressed genes (normalized UMI counts above 5e-6) and zonated genes (normalized UMI counts above 5e-6 zonation $q$-value above 0.2). Pathways were included only if they contained at least 15 expressed genes.

**Deconvolution of the malignant liver LCM**

LCM RNAseq results were deconvolved into the constituting cell types using AutoGeneS version 1.0.3 (preprint: Aliee & Theis, 2020) and non-negative least squares regression. First, cell type-specific gene expression signature vectors were calculated as the centroids of all cells associated with a given cell type cluster after normalization to count sum = 1 per cell. Next, 4,106 genes with high variability across cell types were identified following scanpy's highly_variable_genes method (Wolf *et al*, 2018) (flavor "Seurat", minimum dispersion = 0.3). Next, AuotGeneS was employed to select 700 marker genes out of the pool of highly variable genes (parameters for AutoGeneS method "optimize": ngen = 5000, seed = 0, nfeatures = 700, mode="fixed", offspring_-size = 100, verbose = False). From the set of pareto-optimal solutions returned by AutoGeneS, the solution which minimizes the correlation between cell types was chosen. Finally, samples were

deconvolved using non-negative least squares regression as implemented within AutoGeneS. T-cell cluster splitted into two clusters "non-tumor T cells" and "tumor T cells" based on the annotation of the sample origin.

## Data availability

Data have been deposited in the GenBank GEO database under accession code GSE146409 (http://www.ncbi.nlm.nih.gov/geo/query/acc.cgi?acc = GSE146409). The data can be interactively explored in the web application at the following link: https://itzkovitzwebapps.weizmann.ac.il/webapps/home/. All code is available in the Zenodo repository under the following https://doi.org/10.5281/zenodo.4267877.

**Expanded View** for this article is available online.

## Acknowledgements

We thank Liran Shlush, Itay Tirosh, Ravid Straussman, Lalit Kumar Dubey, Ronen Alon, and Rita Manco for valuable discussions. S.I. is supported by the Wolfson Family Charitable Trust, the Edmond de Rothschild Foundations, the Fannie Sherr Fund, the Helen and Martin Kimmel Institute for Stem Cell Research grant, the Minerva grant, the Israel Science Foundation grant No. 1486/16, the Broad Institute-Israel Science Foundation grant No. 2615/18, the European Research Council (ERC) under the European Union's Horizon 2020 research and innovation program grant No. 768956, the Chan Zuckerberg Initiative grant No. CZF2019-002434, the Bert L. and N. Kuggie Vallee Foundation and the Howard Hughes Medical Institute (HHMI) international research scholar award. This research was supported by the Israeli Ministry Of Science and Technology (MOST) personal grant to H.M.

## Author contributions

SI, HM, and GZ conceived the study. HM designed and performed most of the experiments. HM, KBH, TJ, and MR performed the smFISH experiments. HM performed image analysis. HM and EEM performed LCM experiments. HM, LB, and SI performed the data analysis. AEM and IA assisted with data analysis. SAG, EP, and GZ assisted with sample collection. SI and HM wrote the manuscript with contribution from SAG, EP, and GZ. All of the authors discussed the results and commented on the manuscript.

## Conflict of interest

The authors declare that they have no conflict of interest.

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
