## [Review Process File · Molecular Systems Biology]

A single cell atlas of the human liver tumor microenvironment

Shalev Itzkovitz, Hassan Massalha, Keren Bahar Halpern, Samir Abu Gazala, Tamar Jana, Efi Massasa, Andreas Moor, Lisa Buchauer, Milena Rozenberg, Eli Pikarsky, Ido Amit, and Gideon Zamir
DOI: 10.15252/msb.20209682

Corresponding author(s): Shalev Itzkovitz (shalev.itzkovitz@weizmann.ac.il)

Review Timeline:

Submission Date:	1st May 20
Editorial Decision:	10th Jun 20
Re-submission date:	8th Sep 20
Editorial Decision:	30th Oct 20
Revision Received:	11th Nov 20
Accepted:	18th Nov 20

Editor: Maria Polychronidou

Transaction Report:

Thank you again for submitting your work to Molecular Systems Biology. We have now heard back from the three referees who agreed to evaluate your manuscript. As you will see below, the reviewers raise substantial concerns on your work, which unfortunately preclude its publication in Molecular Systems Biology.

The reviewers appreciate that the presented analyses could be relevant for liver and cancer biology. However, they point out that as it stands the study remains rather preliminary and they are not convinced that it provides sufficient functional insights. The reviewers rated both the novelty and the conclusiveness of the study as Low/Medium and were not supportive of publication of the study in its current form. As such, at this point we see no choice but to return the manuscript with the message that we cannot offer to publish it.

Nevertheless, as the reviewers did acknowledge that the findings seem potentially relevant, we would not be opposed to considering a substantially revised and extended manuscript based on this work, provided that the issues raised by the reviewers can be convincingly addressed. Some of the more essential issues that would need to be addressed include:

- i) As the reviewers point out, the overall advance and significance of the work would be enhanced by including follow up analyses validating some of the findings. Specifically, we agree with the reviewers' recommendations to include descriptions of newly identified markers of cell types, validation of marker expression by orthogonal approaches and follow-up experiments providing functional insights into the predicted ligand-receptor interactions and their implications in cancer.
- ii) Reviewer #2 is concerned that the small number of patients, in combination with the heterogeneity of the samples and the limited follow up validations potentially undermine the relevance of the dataset as a resource. It would be important to perform additional analyses to better support the resource value of the presented datasets.
- iii) Reviewer #2 recommends including validations of potential therapeutic targets (e.g. SLIT-ROBO). We think that these analyses would indeed enhance the impact of the study. However, they are perhaps not mandatory for the acceptance of the study, if the manuscript is instead sufficiently extended by including functional insights into cancer-stroma interactions (point i). That said, we would not be opposed to the inclusion of data on therapeutic targets if you have them at hand and wish to add them to the study.

All three reviewers provide constructive suggestions on how to address the points above and improve the study. All other technical issues would need to be addressed.

Reviewer #1:

The manuscript titled "A single cell atlas of the human liver tumor microenvironment" by Massalha H. et.al, describes the cell composition of malignant and adjacent non-malignant liver, including spatial information and predicted interactions between different cell types. The authors discovered their scRNA-seq data was composed of 17 clusters, which they went on to annotate as carcinoma cells or other cells that make up the TME. Several interesting sub-analyses between the different identified cell subtypes were subsequently done and differentially expressed genes were described for a large portion of the manuscript. Nicely, these comparisons were followed up by predicted ligand-receptor interactions and the incorporation of spatial proximity of cells that may engage in these interactions. Critically, the exact method by which the authors annotated cell types should be clarified to fully understand the novelty of these reported markers genes or if they were in fact used in the process of annotation from the beginning. Novelty of this manuscript could be greatly improved by the description of newly identified markers of cell types, validation of marker expression by smFISH and experimental data analyzing the functional impact of predicted ligand-receptor interactions.

Major points:

The methodology for annotating the 17 clusters of cells found from the scRNA-seq is not clearly stated. In the results section, the authors state that marker genes used for cluster identification came from the Ramachandran et al. 2019 publication. There is no mention of these marker genes being used in the methods section. The methods section states that annotation was based on highly expressed genes from each cluster that generated coarse grain clusters. How did the

authors further delineate from these coarse grain clusters to the specific annotations shown in Figure 1C-D? This is of concern as there is significant overlap of marker genes for each cluster in Table S2. For example, many markers are found in at least 3 of the 4 mesenchymal clusters. The authors should clarify how they distinguished CAFs from stellate cells and pericytes with such dramatic overlap of marker genes.

The stromal-cancer cell interactions described in figure 4B are very interesting. Experimentally inducing these interactions *In vitro* and assessing the impact on cancer cell phenotypes would add greatly to the predictions presented here.

Can any of the interactions in 3E be confirmed using smFISH?

The authors should describe the stromal signatures that are shared between the patients in more detail. What do these signatures describe about the cells besides identifying them as stromal in origin?

Minor points:

The authors should include a tSNE plot overlaying the expression of FGFR2 and CDH17 onto the carcinoma cells in a similar fashion to Figure 3A and 4B.

Do all clusters contain cells from both malignant and non-malignant adjacent tissue? What is the percentage of malignant/non-malignant cells in each cluster?

In figure 1D, carcinoma marker genes are included twice in the figure.

What type of cells are in the cluster marked proliferation? Can the authors elaborate on the nature of the cells in this cluster?

In the paragraph starting at line 99, are the described marker genes included in the Ramachandran et al, 2019 signatures for these cell types? The authors should clarify if these genes were used to annotate the cell types or were found as new marker genes after annotation.

For figure 3C-D, please include the total N for each group in the figure legend.

Please show a representative smFISH staining for PERP as in 5E.

Please include a figure similar to S1B that shows patients as columns and cell types in different colors.

Please include in Figure S2 a heatmap of the average expression for each gene across the different clustered cell types. The stellate cell cluster should also be included.

Reviewer #2:

In the paper by Massalha et al., the authors sequenced by MARS-seq more than 7900 cells from malignant and non-malignant tissues of 6 patients and reconstruct cell-to-cell interactions using

ligand-receptor maps with a special focus on tumor pericyte-endothelial cell and carcinoma-stroma cells interactions. LCM was used to study spatial heterogeneity in one cancer. The paper aims to be the first liver tumor-stroma atlas. Overall, it is well written and cutting-edge techniques were used to conduct the analysis. However, only 6 patients were included, and the studied tissues are very heterogeneous (3 colorectal metastases, 2 intrahepatic cholangiocarcinoma, 1 benign cyst both malignant and non-malignant tissue). This reduces the number of cells per condition and, therefore, the study power and the generalizability of the findings. Because of this limitation, the dataset cannot be considered yet a valuable comprehensive tumor-stroma atlas. The same limitation affects the spatial analysis of tumor heterogeneity which is the weak in terms of relevant findings. Finally, no validation of potential therapeutic target is provided.

I suggest to reject with the possibility to resubmit after major revisions.

MAJOR COMMENTS

- Some of non-malignant cells are annotated as "scar-associated" (e.g. SAMs). More information on underlying liver disease and liver fibrosis in these patients are needed, especially for those with cholangiocarcinoma
- SmFISH in Fig. 3b, why the authors chose PDGFB as endothelial cell marker instead of pan-endothelial markers such as PECAM1?
- Spatial analysis of tumor heterogeneity is not convincing. Cancer is characterized by intertumor and intratumor heterogeneity. The authors mixed 2 patients with cholangiocarcinoma and 1 patient with colon cancer. Moreover, even though the authors normalized for the different prevalence of the carcinoma cells between core and fibrotic areas, the normalization method does not consider the lower cell prevalence (both carcinoma and stromal cells) in the fibrosis. Finally, the relevance of such a small difference in CD24 and PERP expression between core and fibrotic area is not clear. Validation on external patients would be needed.
- No validation of potential therapeutic target (eg. SLIT-ROBO is provided)
- What is the meaning of Huntington, Alzheimer and Parkinson pathways in Figure S3?

MINOR COMMENTS

- Indicate in the legend of Figure 5B the staining performed
- Explain what PERP and RSG5 encode for and their relevance

Reviewer #3:

In this manuscript, the authors collect six liver samples, mostly tumors, and study them using single-cell and spatial gene expression analysis. This manuscript has data that will be important for the single-cell, liver biology, and cancer communities. The focus of the manuscript is on the tumor microenvironment (as mentioned in the title) and cancer is mentioned in every sentence of the abstract. However, many of the presented results are exclusive to general liver biology (Figure 3 and top half of Figure 5). Thus the authors do not make a compelling set of claims for their results, beyond a resource. Below are more specific concerns for this manuscript:

1. It is not clear how the non-malignant samples are extracted from the 5 tumor samples. In Fig. 1A they are shown as two sites; is this how they were collected.
2. In Figure 1D, how is relative expression computed?
3. "FGFR2, which was shown to form gene fusions in cholangiocarcinoma cancer cells", the relationship between the gene fusion and overexpression is unclear.
4. For the Figure 2 analysis, are there particular enrichments of gene ontology terms among the

differentially expressed genes? Also, cholangiocarcinoma and colorectal carcinoma were sampled, and so it is worth testing whether cell types show specific states across these cancer types.

5. "We used smFISH to demonstrate that the RGS5+ cells are indeed adjacent to endothelial cells, marked by PDGFB, as expected from pericytes", it is unclear what this is testing: RGS5 as a marker for pericytes, or the fact the pericytes are near endothelial cells, both of which are known.

6. The Figure 3 analysis does not pertain to cancer. Thus, this appears to be an analysis that does not go farther than identifying putative interactions in a non-cancer framework, and no novel result is claimed.

7. The Figure 4 analysis is not explained clearly. Are the cells from the non-malignant samples used, or only the cells from the malignant samples? It seems that the right analysis would be to identify, as in Figure 2, those genes that are uniquely expressed in CAFs - for example - from tumors relative to non-tumors and then to test if there is evidence for communication with the malignant cells using the receptor-ligand analysis.

8. The authors write "We observed a similar cooperation of immune cells and CAFs within the MET signaling module. MET signaling is a major driver in hepatic tumors and metastases." The evidence for actual cooperation is very weak here since all that was shown was that two populations have related receptors and ligands.

9. The word "malignant" used throughout (for example in Figure 1) should refer to cells, whereas "carcinoma" should refer to samples. In particular, in figure 2A calling them malignant T-cells relative to non-malignant T-cells is very misleading, and would be clearer if replaced by "tumor T-cells" for example.

10. The spatial transcriptomics of the non-malignant liver does not seem to relate to the rest of the paper as it is currently framed around the tumor microenvironment. Furthermore, in the comparison between human and mouse zonation, the authors should make a case for the potential significance of differences and similarities.

Reviewer #1:

The manuscript titled "A single cell atlas of the human liver tumor microenvironment" by Massalha H. et.al, describes the cell composition of malignant and adjacent non-malignant liver, including spatial information and predicted interactions between different cell types. The authors discovered their scRNA-seq data was composed of 17 clusters, which they went on to annotate as carcinoma cells or other cells that make up the TME. Several interesting sub-analyses between the different identified cell subtypes were subsequently done and differentially expressed genes were described for a large portion of the manuscript. Nicely, these comparisons were followed up by predicted ligand-receptor interactions and the incorporation of spatial proximity of cells that may engage in these interactions. Critically, the exact method by which the authors annotated cell types should be clarified to fully understand the novelty of these reported markers genes or if they were in fact used in the process of annotation from the beginning. Novelty of this manuscript could be greatly improved by the description of newly identified markers of cell types, validation of marker expression by smFISH and experimental data analyzing the functional impact of predicted ligand-receptor interactions.

We thank the reviewer for these excellent comments. We have now performed extensive new smFISH validations for a range of our identified cell markers (new supplementary Figure S2). These include the carcinoma marker KRT8, the CAFs markers POSTN, NES and EPHA3, the SAMs markers GPNMB and C1QC, the endothelial marker SOX17, the TM1 marker FCN1, the vSMC marker MCAM, the hepatocytes markers CYP2E1, PCK1, and the malignant cells marker EFNA1.

Figure S2 – smFISH validations of the expression of key cell-type specific markers (A-B) CAF marker POSTN, SAM marker GPNMB and vSMC marker MCAM, (C) Endothelial marker SOX17 and CAFs marker NES, (D) TM1 marker FCN1 and C1QC expressed in SAMs, (E) Hepatocytes markers CYP2E1, PCK1, (F) CAF marker EPHA3 and carcinoma-expressed EFNA1. KRT – pan-keratin antibody staining marking the carcinoma cells. ‘Phall’ in panel E is Phalloidin membrane staining. Scale bar in all panels is 10um

We have now also developed an interaction score, based on our single cell ligand-receptor analysis that can be applied to bulk RNAseq data of different tumors. We applied this score to hundreds of liver tumors from the TCGA database. This score incorporates the joint expression levels of both ligands and their *matching* receptors,

and compares this to randomized scores obtained by degree-preserving network randomizations. This interaction score thus includes information that goes beyond that of the expression of the individual ligands and receptors. We demonstrate that this score strongly correlates with tumor stage (Figure 4B, E-F). This is described on page 11:

“Recurring interaction network connectivity correlates with liver tumor severity

The recurring interactions between the carcinoma cells and cells in the TME suggest that elevated expression of these ligands and their matching receptors could convey a selective advantage to cells in the liver TME. To assess this hypothesis, we examined a large cohort of 383 bulk-sequenced liver tumors from the TCGA database, and computed a network score based on our recurring network connectivity (Fig 4E, F). For each tumor, we first computed a score that consists of the summed products of the expression levels of each ligand and matching receptor, and normalized it by computing a randomized score based on degree-preserving random networks (Methods, Fig 4E). This normalization is important, since a high score may simply reflect elevated expression of the ligands and receptors, often oncogenes, rather than the coordinated expression of ligands and their *matching* receptors. We found that the network score significantly increased along the liver tumor stages (Fig 4F). Thus, our interaction score correlates with tumor severity.”

Figure 4 - Human liver interactome delineates tumor-stroma cross talk

(B) Network of recurring ligand-receptor interactions between carcinoma cells and stromal cells from the malignant sites. Node colors denote the cell type cluster in which the ligands/receptors are enriched. Gray arrows color indicates the interaction Zscore (Methods). Recurring modules are shaded. Included are all recurring interactions that significantly appeared in at least three patients. (E) Strategy for computing interaction scores for each tumor. A score is computed as the sum of the products of all ligands and matching receptors of the recurring interaction network. These are compared to the average score obtained when randomizing the real interaction network in a manner that preserves the number of outgoing and incoming interactions of each ligand and receptor ($Score^{rand}$). The ratio of the real and randomized scores constitutes a network score. (F) Network score increases with increasing tumor stage. Analysis for 383 TCGA sample.

This analysis is also described in the methods section on page 29:

“Network score

A score was computed based on the recurring interactions (Fig 4B) as the sum of the products of the expression of all ligands and matching receptors. Additionally, a randomized score $\text{Score}^{\text{rand}}$ was computed as the mean Score obtained from 1000 degree preserving randomized networks (preserving the number of outgoing and incoming interactions for each ligand and receptor). the randomized networks were generated using mfinder (v1.2) (Milo, 2002) with the command ‘mfinder -r 1000 -ornet -nsr 10’. The network score was defined as the ratio between Score and $\text{Score}^{\text{rand}}$ (Fig 4E). We computed this score on bulk RNAseq samples of primary liver tumors downloaded from the TCGA database. Tumors used included untreated individuals or individuals not reported as treated (383 samples). Kruskalwallis test was conducted to assess the significance of the network score between different tumor stages - tumor stage i, ii, iii (grouped with tumor iii substages), and iv (grouped with tumor substage) (Fig 4E).”

Major points:

The methodology for annotating the 17 clusters of cells found from the scRNA-seq is not clearly stated. In the results section, the authors state that marker genes used for cluster identification came from the Ramachandran et al. 2019 publication. There is no mention of these marker genes being used in the methods section. The methods section states that annotation was based on highly expressed genes from each cluster that generated coarse grain clusters. How did the authors further delineate from these coarse grain clusters to the specific annotations shown in Figure1C-D? This is of concern as there is significant overlap of marker genes for each cluster in Table S2. For example, many markers are found in at least 3 of the 4 mesenchymal clusters. The authors should clarify how they distinguished CAFs from stellate cells and pericytes with such dramatic overlap of marker genes.

We thank the reviewer for this important comment. We have now better explained our methodology and refined the marker detection, as suggested by the reviewer. The clustering and marker identification in Table S2 were performed in an unbiased manner using the Seurat software, by comparing each cluster to all other

cell types. We have now grouped these Seurat clusters into coarse-grain clusters based on the known pan-markers of immune cells (PTPRC=CD45) Endothelial cells (PTPRB), and mesenchymal cells (TAGLN). We have now followed the reviewer's suggestion and have added a new sheet in Table S2 that includes cell-type markers identified with Seurat in comparison to all other cells in the respective *coarse-grained cluster*, in addition to the markers currently used that compared to all cells in the atlas. This is described in page 25:

“Annotation for detected clusters was based on highly expressed genes for each cluster and performed in two stages. In the first stage, markers were detected by comparing each cluster to all other clusters in the atlas (Table S2, general sheet). This resulted in some markers that were found in several clusters (e.g. COL1A1 that marks CAFs, Stellate cells and Pericytes). To further refine the cluster annotation, we grouped the 17 Seurat clusters into coarse-grained clusters based on the known pan-markers of immune cells (PTPRC), endothelial cells (PTPRB), and mesenchymal cells (TAGLN). We then identified coarse-grained marked by comparing each cluster to the other clusters within its coarse-grained group (Table S2, type-specific markers sheet). Carcinoma clusters were merged for downstream analysis. The list of cell type markers (Table S2) was obtained by running the Seurat function ‘FindAllMarkers’ with the following parameters: log-fold change above 0.25, minimum fraction of cells expressing a gene in either of the compared populations is 10%, minimum cells expressing a gene is 10, positive genes only.”

And in the caption of Table S2 (page 20):

“Table S2 - Seurat markers for all cell types

Marker genes of the 17 clusters generated by the Seurat function ‘FindAllMarkers’ (general sheet). Specific markers of cell types belonging to the immune, endothelial and mesenchymal coarse-grained clusters were extracted using Seurat function ‘FindMarkers’. Specific markers of each cell type were tested against the remaining cell types from the same coarse-grained cluster (cell type specific markers sheet). ‘p_val’ – unadjusted p-values, ‘avg_logFC’ - log fold-change in the average expression between the compared cluster and the rest of the cells. ‘pct.1’ and ‘pct.2’ – fraction of positive cells for the marker in the cluster and in the remaining cells respectively, ‘p_val_adj’ –

false discovery rate (FDR) adjusted p-value based on Bonferroni correction. In all compared we retained genes with a minimum log-fold change of 0.25 between the two compared populations, minimum 10 cells in one of the two compared populations, and genes that are detected in a minimum of 10% of cells in either of the two populations.”

We have also added a new supplementary Figure S5A showing some of these distinct markers for each of the four mesenchymal cell types separately.

Figure S5 – Markers of the mesenchymal cell types

(A) Markers for the four mesenchymal cell types, obtained by comparisons between each Seurat cluster and the remaining three clusters in the mesenchymal coarse-grained cluster. Rows denote the mesenchymal cell type (Stellate cells, CAFs, Pericytes, vSMC).

The stromal-cancer cell interactions described in figure 4B are very interesting. Experimentally inducing these interactions In vitro and assessing the impact on cancer cell phenotypes would add greatly to the predictions presented here.

This would have of course been great, however modelling colorectal tumor metastasis interactions with TME components *in-vitro* would be highly variable and non-representative of the real human tumor microenvironment. We agree, however, with these excellent comments regarding the need for extending the functional relevance of our ligand-receptor map and have now done so, by developing the interaction score described above and presented in Figure 4 and applying it to 383 bulk-sequenced liver tumors from the TCGA database. This analysis demonstrates the information content in this kind of analysis, which exceeds that of the expression of the individual ligands and receptors.

Can any of the interactions in 3E be confirmed using smFISH?

We have used smFISH to confirm and validate two of the central interactions we have identified in Figure 3: SLIT2-ROBO4 acting from pericytes to endothelial cells, and PDGFB-PDGFRB acting from endothelial cells to pericytes (Figure S5B,C):

Figure S5

(B-C) smFISH validation of genes mediating the interaction between pericytes and endothelial cells. (B) Tumor pericytes co-expressing RGS5 (magenta dots) and the ligand PDGFRB (red dots), and the endothelial-specific ligand PDGFB (green dots). Dashed line mark endothelial cells that are negative for RGS5. (C) Expression of the ligand SLIT2 mRNAs (red dots) in pericytes

(marked by red dashed lines) and ROBO4 mRNAs (green dots) in endothelial cells (marked by green dashed lines). Blood vessel marked with dashed white line. Dashed box marking the blowup area. Scale bar in all panels is 10um. KRT – pan-keratin immunofluorescence signal highlights the carcinoma cells.

The authors should describe the stromal signatures that are shared between the patients in more detail. What do these signatures describe about the cells besides identifying them as stromal in origin?

We have now elaborated throughout the text on the functional properties of the shared signatures. For example on pages 5-6:

“The immune cell populations in the malignant liver predominantly included scar-associated macrophages (SAMs) (Ramachandran *et al*, 2019) (Fig 2B). These cells express the marker genes CD9 and TREM2, a tumor suppressor in hepatocellular carcinoma (Tang *et al*, 2019), as well as the markers CAPG and GPNMB. GSEA (Subramanian *et al*, 2005) analysis of SAM genes resulted in a significant enrichment of apical junction genes and the complement system. Their recurring signatures included lipid-associated genes, such as PLIN2 and LPL, overlapping the recently identified SPP1+ Lipid Associated Macrophages (LAMs) in mouse fatty livers (Remmerie *et al*, 2020).”

We have also elaborated on the roles of distinct stromal cell types in modulating the ECM and in specifically interacting with the carcinoma cells on pages 9-10 in the section entitled “Recurring interactions between the carcinoma cells and the tumor microenvironment”

Minor points:

The authors should include a tSNE plot overlaying the expression of FGFR2 and CDH17 onto the carcinoma cells in a similar fashion to Figure 3A and 4B.

We have now added this Figure R1 below:

Figure R1 – tSNE plot colored by the expression of the cholangiocarcinoma marker FGFR2 and the metastasis marker CDH17.

While we have not included this image in the manuscript, our new online atlas browser at <https://itzkovitzwebapps.weizmann.ac.il/webapps/home/> enables easily producing the tSNE plots of any gene of interest (Figure R2).

Figure R2 – Dedicated web apps that enable easily exploring our human liver cell atlas and the zonation profiles of hepatocytes (both human and mice).

Do all clusters contain cells from both malignant and non-malignant adjacent tissue?
 What is the percentage of malignant/non-malignant cells in each cluster?

We have now added a new supplementary figure panel showing this analysis (Figure S1C):

Figure S1 – (C) Relative composition of each cluster by sample origin (tumor or non-tumor).

In figure 1D, carcinoma marker genes are included twice in the figure.

We thank the reviewer for noting this, we have now corrected this (Figure 1D).

Figure 1 - (D) Heatmap showing the normalized expression of marker genes for the different clusters (Methods). Expression is normalized by the maximal expression among all cell types.

What type of cells are in the cluster marked proliferation? Can the authors elaborate on the nature of the cells in this cluster?

We have now added a new panel (Figure S1F) demonstrating that the proliferation cluster mainly consists of immune cells.

Figure S1 – (F) Sorted spearman correlations of the mean gene expression of the proliferation cluster and the remaining 16 clusters demonstrate that it consists predominantly of immune cells.

In the paragraph starting at line 99, are the described marker genes included in the Ramachandran et al, 2019 signatures for these cell types? The authors should clarify if these genes were used to annotate the cell types or were found as new marker genes after annotation.

We thank the reviewer for this important point. We have now clarified the markers to highlight the overlap of the mononuclear phagocyte populations we found in the liver TME with the cell populations found by Ramachandran *et. al.* in cirrhotic livers. Specifically, these included the most specific markers described in Figure 2d, e in Ramachandran *et. al.* – CD9 and TREM2 for SAMs, FCN1 and S100A12 for TM1, and MARCO, CD5L, CD163 for Kupffer cells. This is described on pages 5-6: “Differences in TME gene expression between the malignant and non-malignant sites

Our single cell analysis of matching malignant and non-malignant sites within the same patients, enabled identification of gene expression differences in distinct cell populations that compose the TME (Fig 2). Genes elevated in tumor endothelial cells compared to the non-tumor endothelial cells included the von Willebrand factor VWA1, encoding a glycoprotein previously shown to facilitate tumor cell extravasation (Terraube *et al*, 2007), as well as SOX17 (Yang *et al*, 2012) and INSR (Nowak-Sliwinska *et al*, 2019), both shown to promote tumor angiogenesis (Fig 2A). The immune cell populations in the malignant liver predominantly included scar-associated macrophages (SAMs) (Ramachandran *et al*, 2019) (Fig 2B). These cells express the marker genes CD9 and TREM2, a tumor suppressor in hepatocellular carcinoma (Tang *et al*, 2019), as well as the markers CAPG and GPNMB. GSEA (Subramanian *et al*, 2005) analysis of SAM genes resulted in a significant enrichment of apical junction genes and the complement system. Their recurring signatures included lipid-associated genes, such as PLIN2 and LPL, overlapping the recently identified SPP1+ Lipid Associated Macrophages (LAMs) in mouse fatty livers (Remmerie *et al*, 2020). Liver mononuclear phagocyte populations from the non-malignant liver sites were composed of Kupffer cells, expressing C1QB, MARCO, CD5L and CD163 (Fig S4). T cells from the malignant sites were predominantly composed of Tregs, marked by CTLA4 and FOXP3, whereas T cell populations from the non-malignant sites were predominantly composed of cytotoxic T cells, expressing CCL5, GZMK and NKG7 (Fig 2C). These divisions within the immune cell populations suggests a recruitment of immune-suppressive subsets of T cells and macrophages, as previously demonstrated for other tumors (Lambrechts *et al*, 2018; Binnewies *et al*, 2018). Additional immune cell types in the TME included conventional dendritic cells (cDC1 and cDC2), tissue monocytes (TM1), expressing FCN1 and S100A12 (Ramachandran *et al*, 2019) and B cells (Fig 1C, Fig S4).”

To further clarify the annotation of the immune cell types, in the context of the Ramachandran paper, we have added a new Supplementary Figure S4 with the main markers of Ramachandran et al. overlaid on our atlas:

Figure S4 – Immune cell type annotation

tSNE plots of genes used as cell type-specific markers in the cirrhotic liver study by Ramachandran *et al.* Box colors denote the annotation of cell types from Ramachandran *et al.* (Ramachandran *et al.*, 2019)

For figure 3C-D, please include the total N for each group in the figure legend.

We have added this information to the legend of Figure 3 on page 9:

“Figure 3 – Mesenchymal heterogeneity in the liver malignant sites (C-D) Violin plot of the distance from blood vessels of low/high RGS5 expressing cells (n=358 and n= 360 respectively) and low/high COL1A1 expressing cells (n=359 and n= 359 respectively). ‘p’ is the p-value determined by Wilcoxon rank-sum test.”

Please include a figure similar to S1B that shows patients as columns and cell types in different colors.

We have now added a new panel D to Figure S1 with this information:

Figure S1 – Patient distribution across all clusters

(D) Distribution of cell types by patient.

Please include in Figure S2 a heatmap of the average expression for each gene across the different clustered cell types. The stellate cell cluster should also be included.

We have now added a new panel B showing this information (Figure S6B):

Figure S6 – Matrisome analysis

(B) Zscore value of the matrisome in tumor cell types found in the TME. Gene subclasses are color-coded. Stellated cells cluster was added as an out-group. Shown are genes with Zscore>1 or Zscore<-1.

Reviewer #2:

In the paper by Massalha et al., the authors sequenced by MARS-seq more than 7900 cells from malignant and non-malignant tissues of 6 patients and reconstruct cell-to-cell interactions using ligand-receptor maps with a special focus on tumor pericyte-endothelial cell and carcinoma-stroma cells interactions. LCM was used to study spatial heterogeneity in one cancer. The paper aims to be the first liver tumor-stroma atlas. Overall, it is well written and cutting-edge techniques were used to conduct the analysis. However, only 6 patients were included, and the studied tissues are very heterogeneous (3 colorectal metastases, 2 intrahepatic cholangiocarcinoma, 1 benign cyst both malignant and non-malignant tissue). This reduces the number of cells per condition and, therefore, the study power and the generalizability of the findings. Because of this limitation, the dataset cannot be considered yet a valuable comprehensive tumor-stroma atlas. The same limitation affects the spatial analysis of tumor heterogeneity which is the weak in terms of relevant findings. Finally, no validation of potential therapeutic target is provided. I suggest to reject with the possibility to resubmit after major revisions.

We thank the reviewer for these excellent constructive comments. We have now expanded our study in each of these directions. The question of heterogeneity and the ability to faithfully identify recurring gene expression signatures from six patients is a very important comment. A valid question in any cell atlas is whether the cell identities uncovered are sufficiently precisely inferred based on the number of patients. In other words, if I would now add a seventh or eighth patient, would the expression signatures of the cell types we uncovered change? To address this question, we performed a comprehensive analysis of the stability of the expression signatures to the number of patients (new supplementary Figure S3). We sub-sampled all possible sets of 1,2,3,4 and 5 patients out of the complete atlas and compared the correlations of the expression signatures of all cell types with the ones obtained from the complete atlas of 6 patients. This analysis clearly shows that the correlation strongly curtails for all cell types except for the carcinoma cells.

Figure S3 – Stability analysis of the cluster gene expression signatures.

Spearman correlations between the mean gene expression signatures for each of the 19 cell types from sub-samples of the six patients (all possible combinations) to the mean gene expression of the full atlas (all six patients, solid black lines) or to an equally-sized sub-samples of cells from all six patients (dashed red line). Expression signatures converge for most cell types, beyond 3 patients, at 5-6 patients. Inset in carcinoma panel is a magnification of 4-6 patients sub-sampling, highlighting the divergence of the red dashed line the black solid line. The last panel shows the difference between the red dashed line and black line for carcinoma cells (black line) and the remaining TME cell types (gray lines).

This part is described in the text on page 5:

“TME cell types exhibit recurring expression signatures

A common question in single cell analysis is whether the reconstructed cell atlases are stable with regards to the numbers of cells per sample and the numbers of samples (Mereu *et al.*, 2020). This question is particularly important in cancer, due to the profound

levels of inter-patient heterogeneity (Patel *et al*, 2014; Meacham & Morrison, 2013; Alizadeh *et al*, 2015; Marusyk *et al*, 2012). We assessed the stability of the expression signatures obtained from our atlas with regards to the number of sampled patients and the number of sampled cells. To this end, we reconstructed the mean gene expression signatures for each of the 17 cell type clusters, based on sub-samples of the six patients, and equally-sized sub-samples of cells from all patients as controls. We compared these mean expression signatures of subsets of the data with those obtained from the full atlas. We found that the gain in correlations, when adding new patients, strongly curtailed for most cell types beyond 3 patients, and converged on the correlations obtained when sub-sampling cells rather than patients (Fig S3). An exception was the carcinoma cluster, where gene expression signatures changed with each new added patient (Fig S3). Our analysis thus demonstrates that, while carcinoma cells exhibit high inter-patient variability, the liver TME exhibits recurring gene expression signatures that are more uniform between patients.”

We have also generated two new web-browsers enabling easy exploration of our cell atlas and of hepatocyte zonation profiles in human and mouse (described in Figure R3 and available in the online apps <https://itzkovitzwebapps.weizmann.ac.il/webapps/home/> referenced in the paper).

Figure R3 – Dedicated web apps that enable easily exploring our human liver cell atlas and the zonation profiles of hepatocytes (both human and mice).

We have also performed massive new spatial transcriptomics experiments on 63 regions throughout the tumors in four patients (described below). Finally, we developed the network interaction score, which we demonstrate holds information on tumor severity (Figure 4).

Figure 4 - Human liver interactome delineates tumor-stroma cross talk

(B) Network of recurring ligand-receptor interactions between carcinoma cells and stromal cells from the malignant sites. Node colors denote the cell type cluster in which the ligands/receptors are enriched. Gray arrows color indicates the interaction Zscore (Methods). Recurring modules are shaded. Included are all recurring interactions that significantly appeared in at least three patients. **(E)** Strategy for computing interaction scores for each tumor. A score is computed as the sum of the products of all ligands and matching receptors of the recurring interaction network. These are compared to the average score obtained when randomizing the real interaction network in a manner that preserves the number of outgoing and incoming interactions of each ligand and receptor ($Score^{rand}$). The ratio of the real and randomized scores constitutes a network score. **(F)** Network score increases with increasing tumor stage. Analysis for 383 TCGA sample.

MAJOR COMMENTS

- Some of non-malignant cells are annotated as "scar-associated" (e.g. SAMs). More information on underlying liver disease and liver fibrosis in these patients are needed, especially for those with cholangiocarcinoma

This is an important point, we have added this information, the only patient with liver fibrosis was the cholangiocarcinoma patient p2 (Figure R4).

Figure R4 - Mason Trichrome stain substantial fibrosis (blue color), ductal expansion and regions of fatty liver and fibrotic portal nodes in the cholangiocarcinoma patient p2. Scale bar 1mm.

This is now described in the text on page 3:

“The non-malignant sites did not show histological signs of fibrosis, with the exception of the cholangiocarcinoma patient p2 (Methods, Table S1).”

And in the Methods section on page 22:

“The fibrotic status of the non-malignant livers were assessed by a pathologist, using Mason Trichrome stain (Table S1). The only patient that exhibited substantial fibrosis was the cholangiocarcinoma patient p2.”

Importantly, as we show in the new Figure S1C the proportions of non-tumor SAMs is very low.

Figure S1 – (C) Relative composition of each cluster by sample origin (tumor or non-tumor).

- SmFISH in Fig. 3b, why the authors chose PDGFB as endothelial cell marker instead of pan-endothelial markers such as PECAM1?

This is a great question, we considered PECAM1, however we found that PDGFB is in fact a more specific marker for endothelial cells (PECAM1 is also substantially expressed in immune cells, Figure R5). We have now expanded the validations of the pericyte-endothelial cell interactions that include an additional endothelial cell-specific marker – ROBO4. This is shown in Figure S5C.

Figure R5 – tSNE plots of endothelial markers PECAM1, PDGFB, and ROBO4

While we have not included these in the manuscript, our new online atlas browser at <https://itzkovitzwebapps.weizmann.ac.il/webapps/home/> enables easily producing the tSNE plots of any gene of interest (Figures R2,R3).

Figure S5 – Markers of the mesenchymal cell types

(B-C) smFISH validation of genes mediating the interaction between pericytes and endothelial cells. (C) Expression of the ligand SLIT2 mRNAs (red dots) in pericytes (marked by red dashed

lines) and ROBO4 mRNAs (green dots) in endothelial cells (marked by green dashed lines). Blood vessel marked with dashed white line. Dashed box marking the blowup area. Scale bar in all panels is 10um. KRT – pan-keratin immunofluorescence signal highlights the carcinoma cells.

- Spatial analysis of tumor heterogeneity is not convincing. Cancer is characterized by intertumor and intratumor heterogeneity. The authors mixed 2 patients with cholangiocarcinoma and 1 patient with colon cancer. Moreover, even though the authors normalized for the different prevalence of the carcinoma cells between core and fibrotic areas, the normalization method does not consider the lower cell prevalence (both carcinoma and stromal cells) in the fibrosis. Finally, the relevance of such a small difference in CD24 and PERP expression between core and fibrotic area is not clear. Validation on external patients would be needed.

We agree with the reviewer that, unlike the hepatocyte case, where LCM provided ample data for reconstruction of zonation patterns, the inference in the tumor was under-powered. To address this, we have now performed an extensive new experiment that sampled 63 regions from the tumors of 4 patients. In this experiment we refined the spatial analysis to analyze 5 zones within each tumor – tumor border (tb), tumor core (tc), the border between the tumor and the fibrotic regions (ftb), tumor islets within the fibrotic regions (ti) and fibrotic zones (fz). We agree that the analysis of specific carcinoma genes that are differentially expressed could be strongly confound in such measurements, and have therefore removed this analysis, and instead used the new detailed LCM data to identify spatial trends in the relative abundances of the cell types in our atlas among different tumor zones. More specifically, we have performed spatial deconvolution of the individual LCM datasets, using our single cell atlas as a reference, to obtain the proportions of the different cell types in different zones. This analysis, that revealed increased abundance of SAMs and T cells in the tumor border, is presented in Figure 6 on pages 17-18:

Figure 6 – Spatial distribution of the TME populations

(A-B) tissue sections from patients p1 and p3 stained with HistoGene Staining Solution. Laser captured zones are marked with dashed black line including: tumor core (tc), the border between the tumor and the fibrotic regions (ftb), tumor islets within the fibrotic regions (ti) and fibrotic zones (fz) in panel A, and tumor border (tb) in panel B. small images in A are magnifications of the corresponding zones in the large image for the left. White dashed line in B marking the non-malignant liver (nml) border. Scale bars in all panels are 100um. (C) Spatial distribution of the different TME populations across the analyzed five zones. Values are the estimated proportions based on de-convolution of the transcriptome of each zone with the average transcriptome of the cell types (Methods).

And described in the new section on pages 16-17:

“Spatial distributions of TME populations

Solid tumors have been shown to exhibit variability in cell composition and expression programs as a function of the location within the tumor (Keren *et al*, 2018; Giesen *et al*, 2014; Angelo *et al*, 2014; Moncada *et al*, 2018). This heterogeneity is often dictated by cell proximity to spatial landmarks (Halpern *et al*, 2017) that include the tumor boundary, blood vessels (Kumar *et al*, 2019) and immune cell aggregates (Colbeck *et al*, 2017). We next asked whether different TME cell types in our atlas are more abundant in distinct tumor microenvironments.

We used LCM to dissect 63 tissue regions from four patients. These regions included the tumor border (tb), tumor core (tc), the border between the tumor and the fibrotic regions (ftb), tumor islets within the fibrotic regions (ti) and fibrotic zones (fz, Fig 6A,B). We performed bulk RNAseq on these regions (Table S6 ‘malignant liver LCM’ sheet) and used the expression signatures of the cell types in our single cell atlas to estimate the proportions of each cell type, based on the bulk measurements. To this end, we used AutoGeneS (Aliee & Theis, 2020) to deconvolve the data (Fig 6C, Table S9, Methods). We found that different cell types differentially populated different tumor zones. As expected, hepatocytes were most abundant in the tumor border zones and almost completely depleted from other tumor zones. CAFs were most abundant in the fibrotic zones. Pericytes showed spatial abundances that highly overlapped those of LVECT, as expected based on their physical attachments (Fig 3). Notably, SAMs and both tumor T cells and non-tumor T cells showed higher abundance in the tumor border zones (Fig 6C).”

The method used for the deconvolution analysis is explained in the methods section on pages 32-33:

“Deconvolution of the malignant liver LCM

LCM RNAseq results were deconvolved into the constituting cell types using AutoGeneS Version 1.0.3 (Aliee & Theis, 2020) and non-negative least squares regression. First, cell-type specific gene expression signature vectors were calculated as the centroids of all

cells associated with a given cell type cluster after normalization to count sum = 1 per cell. Next, 4106 genes with high variability across cell types were identified following scanpy's highly_variable_genes method (Wolf *et al*, 2018) (flavor 'Seurat', minimum dispersion=0.3). Next, AuotGeneS was employed to select 700 marker genes out of the pool of highly variable genes (parameters for AutoGeneS method 'optimize': ngen=5000, seed=0, nfeatures=700, mode='fixed', offspring_size=100, verbose=False). From the set of pareto-optimal solutions returned by AutoGeneS, the solution which minimizes the correlation between cell types was chosen. Finally, samples were deconvolved using non-negative least squares regression as implemented within AutoGeneS. T cell cluster split into two clusters 'non-tumor T cells' and 'tumor T cells' based on the annotation of the sample origin."

We have also added a new table with the deconvolution analysis. Below is the captions for table S9 (page 22):

"Table S9 – Spatial distribution of TME cells

Deconvolution of the malignant laser captured zones to determine the spatial distributions of the TME cell types."

- No validation of potential therapeutic target (eg. SLIT-ROBO is provided)

We have now used smFISH to validate both SLIT2-ROBO4 acting from pericytes to endothelial cells, and PDGFB-PDGFRB acting from endothelial cells to pericytes (Figure S5B,C):

Figure S5 – Markers of the mesenchymal cell types

(B-C) smFISH validation of genes mediating the interaction between pericytes and endothelial cells. (B) Tumor pericytes co-expressing RGS5 (magenta dots) and the ligand PDGFRB (red dots), and the endothelial-specific ligand PDGFB (green dots). Dashed line mark endothelial cells that are negative for RGS5. (C) Expression of the ligand SLIT2 mRNAs (red dots) in pericytes (marked by red dashed lines) and ROBO4 mRNAs (green dots) in endothelial cells (marked by green dashed lines). Blood vessel marked with dashed white line. Dashed box marking the blowup area. Scale bar in all panels is 10um. KRT – pan-keratin immunofluorescence signal highlights the carcinoma cells.

- What is the meaning of Huntington, Alzheimer and Parkinson pathways in Figure S3?

These are part of the GESA output, while not functionally coherent, we decided to leave them in to avoid exclusion bias. We added a new Supplementary Table S8 detailing the genes in each of the gene sets we found to be enriched in Figure 5.

Table S8 caption (page 22):

“Table S8 – Genes list of hepatocytes enriched pathways

List of genes of the GSEA enriched pathways in human hepatocytes.”

MINOR COMMENTS

- Indicate in the legend of Figure 5B the staining performed

We have added this information to the legend of Figure 5B on pages 15-16:

“(B) Non-malignant human liver lobule from patient p4 used for LCM stained with HistoGene Staining Solution (Methods). Top – sequential zones from the central vein (cv) to the portal node (pn) before microdissection. LCM captured zones marked by the black dashed line. Bottom – Tissue after microdissection. Scale bar is 100µm.”

- Explain what PERP and RSG5 encode for and their relevance

As explained above we have now removed this section.

Reviewer #3:

In this manuscript, the authors collect six liver samples, mostly tumors, and study them using single-cell and spatial gene expression analysis. This manuscript has data that will be important for the single-cell, liver biology, and cancer communities. The focus of the manuscript is on the tumor microenvironment (as mentioned in the title) and cancer is mentioned in every sentence of the abstract. However, many of the presented results are exclusive to general liver biology (Figure 3 and top half of Figure 5). Thus the authors do not make a compelling set of claims for their results, beyond a resource. Below are more specific concerns for this manuscript:

We are sorry for the confusion, the entire analysis of Figure 3 pertains to the tumor region, where the pericytes were abundant. There are practically no pericytes in the non-malignant human liver. We now elaborate on this and have added a new figure panel showing this (Figure S1C). Our new analyses described above, including smFISH validations, application of the interaction scores to hundreds of liver tumors, new spatial analyses and new web-browsers for exploring the liver tumor cell atlas significantly expands the potential impact of this study.

1. It is not clear how the non-malignant samples are extracted from the 5 tumor samples. In Fig. 1A they are shown as two sites; is this how they were collected.

We have now explained this in the methods section on page 22:

“In each surgery, two distinct samples were obtained – one from the tumor region (‘malignant site’) and another from an adjacent non-tumor region that was at least 1-5cm from the tumor region (‘non-malignant site’).”

2. In Figure 1D, how is relative expression computed?

We have now explained this in the caption and updated the panel D on pages 4-5:

“(D) Heatmap showing the normalized expression of marker genes for the different clusters (Methods). Expression is normalized by the maximal expression among all cell types.”

3. "FGFR2, which was shown to form gene fusions in cholangiocarcinoma cancer cells", the relationship between the gene fusion and overexpression is unclear.

We agree and have deleted the statement regarding the gene fusion.

4. For the Figure 2 analysis, are there particular enrichments of gene ontology terms among the differentially expressed genes? Also, cholangiocarcinoma and colorectal carcinoma were sampled, and so it is worth testing whether cell types show specific states across these cancer types.

This is an excellent idea, we have now added a new figure panel (Figure 2D) with this analysis. The only TME cell population that exhibited changes between the cholangiocarcinoma and metastasis patients were the mononuclear phagocytes. This is described on page 7:

“We further assessed the differences in the expression signatures of endothelial cells, mononuclear phagocytes and T cells between the tumor sites of the cholangiocarcinoma patients and the metastases patients (Fig 2D). Endothelial cells and T cells did not exhibit differential expression between these two etiologies. In contrast, mononuclear phagocytes exhibited up-regulation of the chemokines such as CCL4, CCL4L2, CCL3L3 in the cholangiocarcinoma samples and extracellular remodeling genes such as MMP19, MMP12 and HS3ST2 in the metastatic patients.”

Figure 2 – (D) DGE analysis between tumor mononuclear phagocytes classified by cancer type (cholangiocarcinoma in dark purple and metastasis in light purple).

5. "We used smFISH to demonstrate that the RGS5+ cells are indeed adjacent to endothelial cells, marked by PDGFB, as expected from pericytes", it is unclear what this is testing: RGS5 as a marker for pericytes, or the fact the pericytes are near endothelial cells, both of which are known.

Our study is the first study to describe the expression signatures of pericytes in the malignant human liver. Indeed, RGS5 was shown to be a marker of pericytes in some studies and, per definition, pericytes are attached to endothelial cells, however it was important to demonstrate that in the context of the human liver tumor microenvironment RGS5 is indeed a specific marker of pericytes, which we demonstrated with the smFISH experiments. This is especially important, since we found other genes that were previously suggested to be markers of pericytes in other contexts, such as DES and ANPEP, to be marking other cells in the human liver TME (Figure R6 below). We now describe this in the text on page 8:

“We found that some previously suggested markers of pericytes, such as DES (Nehls *et al*, 1992) and ANPEP (Kumar *et al*, 2017) were not specifically expressed in pericytes in

the malignant human liver context (Table S3). Importantly, Pericytes were almost absent from the non-malignant sites (Fig S1C).”

Figure R5 – tSNE plots of two previously documented pericytes markers, ANPEP and DES, found to be expressed in other cell types (LSEC and cholangiocytes for ANPEP, vSMCs for DES) but not in pericytes. tSNE plot of the pericyte marker RGS5 is shown in the most right panel.

6. The Figure 3 analysis does not pertain to cancer. Thus, this appears to be an analysis that does not go farther than identifying putative interactions in a non-cancer framework, and no novel result is claimed.

We are sorry that we have not clarified this section sufficiently, all of Figure 3 pertains to the malignant sites, where the pericytes are found. There are essentially no pericytes in the non-malignant liver. We now show this in the new panel Figure S1C and have better stressed this in the text on page 8:

“Importantly, Pericytes were almost absent from the non-malignant sites (Fig S1C).”

The added panel is shown below in Figure S1C.

Figure S1 – (C) Relative composition of each cluster by sample origin (tumor or non-tumor).

We have now also delved deeper to validate in-situ the interactions we identified between tumor pericytes and tumor endothelial cells, by imaging smFISH libraries for SLIT2 and ROBO4, PDGFB and PDGFRB (Figure S5B,C)

Figure S5 – Markers of the mesenchymal cell types

(B-C) smFISH validation of genes mediating the interaction between pericytes and endothelial cells. (B) Tumor pericytes co-expressing RGS5 (magenta dots) and the ligand PDGFRB (red dots), and the endothelial-specific ligand PDGFB (green dots). Dashed line mark endothelial cells that are negative for RGS5. (C) Expression of the ligand SLIT2 mRNAs (red dots) in pericytes (marked by red dashed lines) and ROBO4 mRNAs (green dots) in endothelial cells (marked by green dashed lines). Blood vessel marked with dashed white line. Dashed box marking the blowup area. Scale bar in all panels is 10µm. KRT – pan-keratin immunofluorescence signal highlights the carcinoma cells.

7. The Figure 4 analysis is not explained clearly. Are the cells from the non-malignant samples used, or only the cells from the malignant samples? It seems that the right analysis would be to identify, as in Figure 2, those genes that are uniquely expressed in CAFs - for example - from tumors relative to non-tumors and then to test if there is evidence for communication with the malignant cells using the receptor-ligand analysis.

Indeed, this is precisely the analysis we have performed in Figure 4, which analyzes the recurring interactions between carcinoma cells and the TME cells in the malignant regions. We have now explained this better in the text on page 10:

“To this end, we parsed a database of ligand-receptor interactions (Ramilowski *et al.*, 2015) and identified pairs, for which the interacting proteins were specific to the carcinoma cell cluster on the one hand, and to the supporting stromal cell clusters on the other (Zhou *et al.*, 2017; Halpern *et al.*, 2018) (Methods).”

8. The authors write "We observed a similar cooperation of immune cells and CAFs within the MET signaling module. MET signaling is a major driver in hepatic tumors and metastases." The evidence for actual cooperation is very weak here since all that was shown was that two populations have related receptors and ligands.

We thank the reviewer for pointing this out, we have now performed a new analysis that uses our recurring interaction network that demonstrates that the interaction itself, rather than the expression levels of the ligands and receptors, correlate with tumor severity stage. This analysis is described on page 11

“Recurring interaction network connectivity correlates with liver tumor severity

The recurring interactions between the carcinoma cells and cells in the TME suggest that elevated expression of these ligands and their matching receptors could convey a selective advantage to cells in the liver TME. To assess this hypothesis, we examined a large cohort of 383 bulk-sequenced liver tumors from the TCGA database, and computed a network score based on our recurring network connectivity (Fig 4E, F). For each tumor,

we first computed a score that consists of the summed product of the expression levels of each ligand and matching receptor, and normalized it by computing a randomized score based on degree-preserving random networks (Methods, Fig 4E). This normalization is important, since a high score may simply reflect elevated expression of the ligands and receptors, often oncogenes, rather than the coordinated expression of ligands and their matching receptors. We found that the network score significantly increased along the liver tumor stages (Fig 4F). Thus, our interaction score correlates with tumor stage.”

The results of this analysis are shown in Figure 4E-F):

Figure 4 - Human liver interactome delineates tumor-stroma cross talk

(B) Network of recurring ligand-receptor interactions between carcinoma cells and stromal cells from the malignant sites. Node colors denote the cell type cluster in which the ligands/receptors are enriched. Gray arrows color indicates the interaction Zscore (Methods). Recurring modules are shaded. Included are all recurring interactions that significantly appeared in at least three patients. (E) Strategy for computing interaction scores for each tumor. A score is computed as the sum of the products of all ligands and matching receptors of the recurring interaction network. These are compared to the average score obtained when randomizing the real interaction network in a manner that preserves the number of outgoing and incoming interactions of each ligand and receptor ($\text{Score}^{\text{rand}}$). The ratio of the real and randomized scores constitutes a network score. (F) Network score increases with increasing tumor stage. Analysis for 383 TCGA sample.

And explained in the Methods part on page 29:

“Network score

A score was computed based on the recurring interactions (Fig 4B) as the sum of the products of the expression of all ligands and matching receptors. Additionally, a randomized score $\text{Score}^{\text{rand}}$ was computed as the mean Score obtained from 1000 degree preserving randomized networks (preserving the number of outgoing and incoming interactions for each ligand and receptor). the randomized networks were generated using mfinder (v1.2) (Milo, 2002) with the command ‘mfinder -r 1000 -ornet -nsr 10’. The network score was defined as the ratio between Score and $\text{Score}^{\text{rand}}$ (Fig 4E). We computed this score on bulk RNAseq samples of primary liver tumors downloaded from the TCGA database. Tumors used included untreated individuals or individuals not reported as treated (383 samples). Kruskalwallis test was conducted to assess the significance of the network score between different tumor stages (tumor stage i, ii, iii (grouped with tumor iii substages), and iv (grouped with tumor substage)) (Fig 4E).”

9. The word "malignant" used throughout (for example in Figure 1) should refer to cells, whereas "carcinoma" should refer to samples. In particular, in figure 2A calling them malignant T-cells relative to non-malignant T-cells is very misleading, and would be clearer if replaced by "tumor T-cells" for example.

We completely agree and have replaced all occurrences of references to ‘malignant cells’ that are not carcinoma (e.g. ‘malignant T cells’) with ‘tumor’ (e.g. ‘Tumor T cells’).

10. The spatial transcriptomics of the non-malignant liver does not seem to relate to the rest of the paper as it is currently framed around the tumor microenvironment. Furthermore, in the comparison between human and mouse zonation, the authors should make a case for the potential significance of differences and similarities.

We thank the reviewer for this comment, we have now significantly elaborated on the zonation analysis of hepatocytes. We have already identified an intriguing discordance between the zonation of fatty acid metabolic enzymes in mouse and human. While lipogenesis genes such as Fasn and Acly are periportally zoned in mouse livers, they are pericentrally zoned in human liver. We have now elaborated on additional concordant and discordant genes and pathways between mouse and human. This is described in the text on pages 14-15:

“We found that, as observed in mice, many hepatocyte genes were significantly zoned along the lobule radial axis (2677 genes out of 8536 genes with expression higher than $1e-5$ of cellular UMIs had $q\text{-value} < 0.2$, Fig 5C, Table S7, ‘reconstructed_human_hepatocytes’ sheet). Pericentrally zoned processes included primary bile acid biosynthesis and metabolism of xenobiotics, whereas periportally zoned processes included oxidative phosphorylation and fructose and mannose metabolism (Fig 5D, Fig S9). We compared the zonation profiles in mouse (Halpern *et al*, 2017) and human and found both discordant and concordant profiles (Table S7, ‘human_mouse_comparison’ sheet, Fig 5D,E, Fig S9). Genes that exhibited overlapping profiles between mice and human included the pericentral genes CYP1A2 and LGR5 and the periportal genes HAL and SDS. The gene SLC2A2, encoding the main hepatic glucose transporter GLUT2 exhibited pericentral zonation, in contrast to its periportal zonation in mice. Similarly, the lipogenesis genes SREBF1, ACLY, FASN and ACACA were pericentrally-zoned in human and periportally-zoned in mice (Table S7, ‘human_mouse_comparison’ sheet, Fig 5E). Additional discordant pathways included

ribosomes, which were periportally zoned in human and pericentrally zoned in mouse, and complement and coagulation cascades, which were periportally zoned in mouse and pericentrally zoned in human (Fig 5E, Fig S9).”

We have now added two new figure panels – Figures 5D, E that highlight common and diverging zoned pathways and genes between human and mice.

Figure 5 – (D) Gene set enrichment analysis of genes with maximal expression above $1e^{-5}$ normalized UMI counts. Centrally (cv) enriched sets are marked with blue dots, portally (pn) enriched sets are marked with red dots. Green fonts denote pathways that are concordantly zoned in mouse, black fonts denote pathways that are not zoned in mouse and red fonts denote pathways that are inversely zoned in mouse. (E) Zonation profiles of selected genes from human (blue) and mouse (green). Each profile is normalized by the maximal expression across zones. Patches are standard errors of the mean. Mouse data used in D and E was obtained from Halpern et al. (Halpern *et al.*, 2017).

We have also added a new supplementary Figure S9 that highlights similarly zoned and differentially zoned KEGG pathways in mouse and human.

Figure S9 – Comparison of Kegg pathways enrichment analysis between human and mouse
 (A) Kegg pathway enriched in zoned genes in both human and mouse. Pathways are sorted by the centers of mass of the average zonation profile in mouse (Halpern *et al.*, 2017). (B-C) Kegg pathway enriched in zoned genes in mouse but not in human (B) and human but not mouse (C). Expression values are normalized to the maximum expression across the layers. Pathways profile are sorted by the center of mass from the central vein (cv) to the portal node (pn).

Finally, we have added a new online web application (<https://itzkovitzwebapps.weizmann.ac.il/webapps/home/>) that enables easy exploration of the zonation patterns in mouse and human for any gene/genes of interest (Figure R7).

Figure R7 – Dedicated web apps that enable easily exploring our human liver cell atlas and the zonation profiles of hepatocytes (both human and mice).

Thank you again for submitting your work to Molecular Systems Biology. We have now heard back from the two referees who agreed to evaluate your study. As you will see below, the reviewers are satisfied with the modifications made and are supportive of publication. Reviewer #2 raises a couple of minor concerns, which we would ask you to address in a revision.

On a more editorial level, we would ask you to address the following.

REFEREE REPORTS

Reviewer #1:

The authors have addressed all my comments and I don't have any further comments about the manuscript. I think the manuscript is acceptable.

Reviewer #2:

With this revision, the authors addressed several of the major concerns regarding the study. It is a descriptive study on cell prevalence and interactions. I appreciated the new LCM analysis provided in Figure 6. The normal human liver zonation analysis has little novelty and is disconnected from the rest since it is not needed for the spatial transcriptomic analysis of the tumor but I appreciated the online tool to explore liver zonation. I suggest to accept the paper with minor revisions.

Minor comment:

- Figure 4F: This panel, which shows small differences stages, represents the network score only for primary liver cancers, thus, colon cancer metastasis are excluded (which are included in the atlas). Please indicate in the legend which kind of tumors were extracted from TCGA and use capital roman numerals for stages.
- Typo "Mason Trichrome" instead of "Masson Trichrome".

Reviewer #1:

The authors have addressed all my comments and I don't have any further comments about the manuscript. I think the manuscript is acceptable.

Reviewer #2:

With this revision, the authors addressed several of the major concerns regarding the study. It is a descriptive study on cell prevalence and interactions. I appreciated the new LCM analysis provided in Figure 6. The normal human liver zonation analysis has little novelty and is disconnected from the rest since it is not needed for the spatial transcriptomic analysis of the tumor but I appreciated the online tool to explore liver zonation. I suggest to accept the paper with minor revisions.

Minor comment:

- Figure 4F: This panel, which shows small differences stages, represents the network score only for primary liver cancers, thus, colon cancer metastasis are excluded (which are included in the atlas). Please indicate in the legend which kind of tumors were extracted from TCGA and use capital roman numerals for stages.

We thank the reviewer for this comment. Tumors type included in this analysis are Hepatocellular Carcinoma (LIHC) and Cholangiocarcinoma (CHOL) and were add to the legend of figure 4F (line 288 - 289): “(F) Network score increases with increasing tumor stage. Analysis for 383 TCGA sample of Liver Hepatocellular Carcinoma (LIHC) and Cholangiocarcinoma (CHOL).”

Stage labels were corrected in figure 4F

- Typo "Mason Trichrome" instead of "Masson Trichrome".

We thank the reviewer for noticing this typo which was corrected (page 22 line 532).

Thank you again for sending us your revised manuscript and for performing the last requested edits. We are now satisfied with the modifications made and I am pleased to inform you that your paper has been accepted for publication.

Corresponding Author Name: Prof. Shalev Itzkovitz

Manuscript Number: MSB-20-9682R